# Exclusively Penalized Q-learning for Offline Reinforcement Learning

**Junghyuk Yeom**\*    **Yonghyeon Jo**\*    **Jungmo Kim**    **Sanghyeon Lee**    **Seungyul Han**†
Graduate School of Artificial Intelligence
UNIST
Ulsan, South Korea 44919
{junghyukyum,yonghyeonjo,jmkim22,sanghyeon,syhan}@unist.ac.kr

## Abstract

Constraint-based offline reinforcement learning (RL) involves policy constraints or imposing penalties on the value function to mitigate overestimation errors caused by distributional shift. This paper focuses on a limitation in existing offline RL methods with penalized value function, indicating the potential for underestimation bias due to unnecessary bias introduced in the value function. To address this concern, we propose Exclusively Penalized Q-learning (EPQ), which reduces estimation bias in the value function by selectively penalizing states that are prone to inducing estimation errors. Numerical results show that our method significantly reduces underestimation bias and improves performance in various offline control tasks compared to other offline RL methods.

## 1   Introduction

Reinforcement learning (RL) is gaining significant attention for solving complex Markov decision process (MDP) tasks. Traditionally, online RL develops advanced decision-making strategies through continuous interaction with environments [1, 2, 3, 4, 5, 6]. However, in real-world scenarios, interacting with the environment can be costly, particularly in high-risk environments like disaster situations, where obtaining sufficient data for learning is challenging [7, 8]. In such setups, the need for exploration [9, 10, 11, 12] to discover optimal strategies often incurs additional costs, as agents must try various actions, some of which may be inefficient or risky [13, 14]. This highlights the significance of research on offline setups, where policies are learned using pre-collected data without any direct interaction with the environment [15, 16]. In offline setups, policy actions not present in the data may introduce extrapolation errors, disrupting accurate value estimation by causing a large overestimation error in the value function, known as the distributional shift problem [17].

To address the distributional shift problem, Fujimoto et al. [17] proposes batch-constrained $Q$-learning (BCQ), assuming that policy actions are selected from the dataset only. Ensuring optimal convergence of both the policy and value function under batch-constrained RL setups [17], BCQ demonstrates stable learning in offline setups and outperforms behavior cloning (BC) techniques [18], which simply mimic actions from the dataset. However, the policy constraint of BCQ strongly limits the policy space, prompting further research to find improved policies by relaxing constraints based on the support of the policy using metrics like maximum mean discrepancy (MMD) [19] or Kullback–Leibler (KL) divergence [20]. While these methods moderately relax policy restrictions, the issue of limited policies persists. Thus, instead of constraining the policy space directly, alternative offline RL methods have been proposed to reduce overestimation bias based on penalized $Q$-functions [21, 22]. Conservative $Q$-learning (CQL) [21], a representative offline RL algorithm using $Q$-penalty, penalizes

---

\* indicates equal contribution and † indicates the corresponding author: Seungyul Han.
Special thanks to Whiyoung Jung from LG AI Research for providing experimental data used in this work.

the $Q$-function for policy actions and provides a bonus to the $Q$-function for actions in the dataset. Consequently, CQL selects more actions from the dataset, effectively reducing overestimation errors without policy constraints.

While CQL has demonstrated outstanding performance across various offline tasks, we observed that it introduces unnecessary estimation bias in the value function for states that do not contribute to overestimation. This issue becomes more pronounced as the level of penalty increases, resulting in performance degradation. To address this issue, this paper introduces a novel Exclusively Penalized Q-learning (EPQ) method for efficient offline RL. EPQ imposes a threshold-based penalty on the value function exclusively for states causing estimation errors to mitigate overestimation bias without introducing unnecessary bias in offline learning. Experimental results demonstrate that our proposed method effectively reduces both overestimation bias due to distributional shift and underestimation bias due to the penalty, allowing a more accurate evaluation of the current policy compared to the existing methods. Numerical results reveal that EPQ significantly outperforms other state-of-the-art offline RL algorithms on various D4RL tasks [23].

## 2 Preliminaries

### 2.1 Markov Decision Process and Offline RL

We consider a Markov Decision Process (MDP) environment denoted as $\mathcal{M} := (\mathcal{S}, \mathcal{A}, P, R, \gamma)$, where $\mathcal{S}$ is the state space, $\mathcal{A}$ is the action space, $P$ represents the transition probability, $\gamma$ is the discount factor, and $R$ is the bounded reward function. In offline RL, transition samples $d_t = (s_t, a_t, r_t, s_{t+1})$ are generated by a behavior policy $\beta$ and stored in the dataset $D$. We can empirically estimate $\beta$ as $\hat{\beta}(a|s) = \frac{N(s,a)}{N(s)}$, where $N$ represents the number of data points in $D$. We assume that $\mathbb{E}_{s \sim D, a \sim \beta}[f(s,a)] \approx \mathbb{E}_{s \sim D, a \sim \hat{\beta}}[f(s,a)] = \mathbb{E}_{s,a \sim D}[f(s,a)]$ for arbitrary function $f$. Utilizing only the provided dataset without interacting with the environment, our objective is to find a target policy $\pi$ that maximizes the expected discounted return, denoted as $J(\pi) := \mathbb{E}_{s_0, a_0, s_1, \cdots \sim \pi}[G_0]$, where $G_t = \sum_{l=t}^{\infty} \gamma^{l-t} R(s_l, a_l)$ represents the discounted return.

### 2.2 Distributional Shift Problem in Offline RL

In online RL, the optimal policy that maximizes $J(\pi)$ is found through iterative policy evaluation and policy improvement [2, 3]. For policy evaluation, the action value function is defined as $Q^\pi(s_t, a_t) := \mathbb{E}_{s_t, a_t, s_{t+1}, \cdots \sim \pi}[\sum_{l=t}^{\infty} \gamma^{l-t} R(s_l, a_l)|s_t, a_t]$. $Q^\pi$ can be estimated by iteratively applying the Bellman operator $\mathcal{B}^\pi$ to an arbitrary $Q$-function, where $(\mathcal{B}^\pi Q)(s, a) := R(s,a) + \gamma \mathbb{E}_{s' \sim P(\cdot|s,a), \, a' \sim \pi(\cdot|s')}[Q(s', a')]$. The $Q$-function is updated to minimize the Bellman error using the dataset $D$, given by $\mathbb{E}_{s,a \sim D}\left[(Q(s,a) - \mathcal{B}^\pi Q(s,a))^2\right]$. In offline RL, samples are generated by the behavior policy $\beta$ only, resulting in estimation errors in the $Q$-function for policy actions not present in the dataset $D$. The policy $\pi$ is updated to maximize the $Q$-function, incorporating the estimation error in the policy improvement step. This process accumulates positive bias in the $Q$-function as iterations progress [17].

### 2.3 Conservative $Q$-learning

To mitigate overestimation in offline RL, conservative Q-learning (CQL) [21] penalizes the $Q$-function for the policy actions $a \sim \pi$ and increases the $Q$-function for the data actions $a \sim \hat{\beta}$ while minimizing the Bellman error, where the $Q$-loss function of CQL is given by

$$\frac{1}{2}\mathbb{E}_{s,a,s' \sim D}\left[(Q(s,a) - \mathcal{B}^\pi Q(s,a))^2\right] + \alpha \mathbb{E}_{s \sim D}[\mathbb{E}_{a \sim \pi}[Q(s,a)] - \mathbb{E}_{a \sim \hat{\beta}}[Q(s,a)]], \quad (1)$$

where $\alpha \geq 0$ is a penalizing constant. From the value update in (1), the average $Q$-value of data actions $\mathbb{E}_{a \sim \hat{\beta}}[Q(s,a)]$ becomes larger than the average $Q$-value of target policy actions $\mathbb{E}_{a \sim \pi}[Q(s,a)]$ as $\alpha$ increases. As a result, the policy will tend to choose the data actions more from the policy improvement step, effectively reducing overestimation error in the $Q$-function [21].

## 3 Methodology

### 3.1 Motivation: Necessity of Mitigating Unnecessary Estimation Bias

In this section, we focus on the penalization behavior of CQL, one of the most representative penalty-based offline RL methods, and present an illustrative example to show that unnecessary estimation bias can occur in the $Q$-function due to the penalization. As explained in Section 2.3, CQL penalizes the $Q$-function for policy actions and increases the $Q$-function for data actions in (1). When examining the $Q$-function for each state-action pair $(s, a)$, the $Q$-value increases if $\pi(a|s) > \hat{\beta}(a|s)$; otherwise, the $Q$-value decreases as the penalizing constant $\alpha$ becomes sufficiently large [21].

To visually demonstrate this, Fig. 1 depicts histograms of the fixed policy $\pi$ and the estimated behavior policy $\hat{\beta}$ for various $\pi$ and $\beta$ at the initial state $s_0$ on the Pendulum task with a single-dimensional action space in OpenAI Gym tasks [24], as cases (a), (b), and (c), along with the estimation bias in the $Q$-function for CQL with various penalizing factors $\alpha$. In this example, for all states except the initial state, we consider $\pi = \beta = \text{Unif}(-2, 2)$. In each case, CQL only updates the $Q$-function with its penalty to evaluate $\pi$ in an offline setup, as shown in equation (1), and we plot the estimation bias of CQL, which represents the average difference between the learned $Q$-function and the expected return $G_0$.

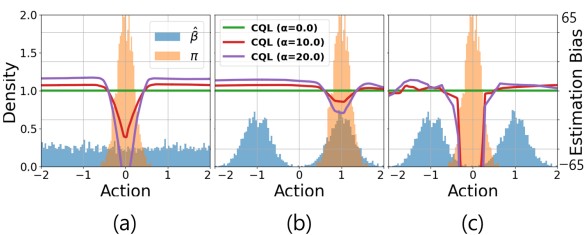

Figure 1: Histograms of $\pi$ and $\hat{\beta}$ (left axis), and the estimation bias of CQL with various $\alpha$ (right axis) at $s_0$ for three cases: (a) $\beta = \text{Unif}(-2, 2)$ and $\pi = N(0, 0.2)$ (b) $\beta = \frac{1}{2}N(-1, 0.3) + \frac{1}{2}N(1, 0.3)$ and $\pi = N(1, 0.2)$ (c) $\beta = \frac{1}{2}N(-1, 0.3) + \frac{1}{2}N(1, 0.3)$ and $\pi = N(0, 0.2)$, where $\text{Unif}(-2, 2)$ represents a uniform distribution and $N(\mu, \sigma)$ denotes a Gaussian distribution with mean $\mu$ and standard deviation $\sigma$.

From the results in Fig. 1, we observe that CQL suffers from unnecessary estimation bias in the $Q$-function for cases (a) and (b). In both cases, the histograms illustrate that policy actions are fully contained in the dataset $\hat{\beta}$, suggesting that the estimation error in the Bellman update is unlikely to occur even without any penalty. However, CQL introduces a substantial negative bias for actions near 0 where $\pi(0|s_0) > \hat{\beta}(0|s_0)$ and a positive bias for other actions. Furthermore, the bias intensifies as the penalty level $\alpha$ increases. In order to mitigate this bias, reducing the penalty level $\alpha$ to zero may seem intuitive in cases like Fig. 1(a) and Fig. 1(b). However, such an approach would be inadequate in cases like Fig. 1(c). In this case, because policy actions close to 0 are rare in the dataset, penalization is necessary to address overestimation caused by estimation errors in offline learning. Furthermore, this problem may become more severe in actual offline learning situations, as the policy continues to change as learning progresses, compared to situations where a fixed policy is assumed.

### 3.2 Exclusively Penalized Q-learning

To address the issue outlined in Section 3.1, our goal is to selectively give a penalty to the $Q$-function in cases like Fig. 1(c), where policy actions are insufficient in the dataset while minimizing unnecessary bias due to the penalty in scenarios like Fig. 1(a) and Fig. 1(b), where policy actions are sufficient in the dataset. To achieve this goal, we introduce a novel exclusive penalty $\mathcal{P}_\tau$ defined by

$$\mathcal{P}_\tau := \underbrace{f_\tau^{\pi,\hat{\beta}}(s)}_{\text{penalty adaptation factor}} \cdot \underbrace{\left( \frac{\pi(a|s)}{\hat{\beta}(a|s)} - 1 \right)}_{\text{penalty term}}, \tag{2}$$

where $f_\tau^{\pi,\hat{\beta}}(s) = \mathbb{E}_{a \sim \pi(\cdot|s)}[x_\tau^{\hat{\beta}}]$ is a penalty adaptation factor for a given $\hat{\beta}$ and policy $\pi$. Here, $x_\tau^{\hat{\beta}} = \min(1.0, \exp(-(\log \hat{\beta}(a|s) - \tau)))$ represents the amount of adaptive penalty that is reduced as $\log \hat{\beta}$ exceeds the threshold $\tau$. Thus, the adaptation factor $f_\tau^{\pi,\hat{\beta}}$ indicates the average penalty that policy actions should receive. If the probability of estimated behavior policy $\hat{\beta}$ for policy actions exceeds the threshold $\tau$, i.e., policy actions are sufficiently present in the dataset, then $x_\tau^{\hat{\beta}}$ will be

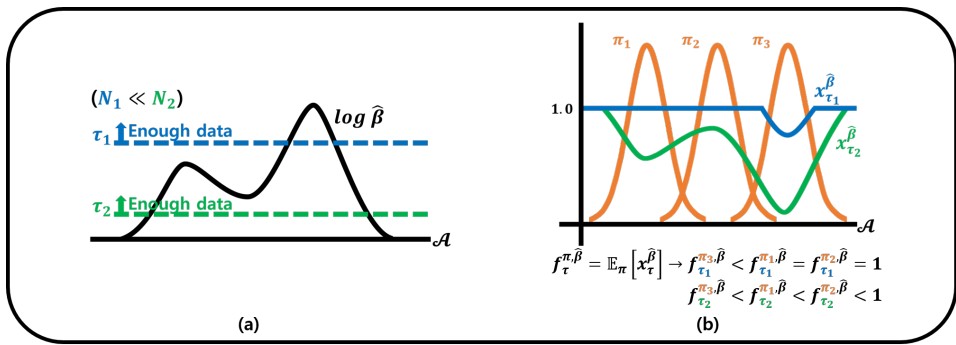

Figure 2: An illustration of our exclusive penalty: (a) The log-probability of $\hat{\beta}$ and the thresholds $\tau_1$ and $\tau_2$ according to the number of data samples $N_1$ and $N_2$, where $N_1 << N_2$. (b) The penalty adaptation factor $f_\tau^{\pi,\hat{\beta}}$ which represents the amount of adaptive penalty, indicating how much $\log\hat{\beta}$ exceeds the threshold $\tau$. Three different policies $\pi_i$, $i = 1, 2, 3$, are considered.

smaller than 1 and reduce the amount of penalty as much as the amount by which $\hat{\beta}$ exceeds the threshold $\tau$ to avoid unnecessary bias introduced in Section 3.1. Otherwise, it will be 1 due to $\min(1.0, \cdot)$ to maintain the penalty since policy actions are insufficient in the dataset. The latter penalty term $\frac{\pi(a|s)}{\hat{\beta}(a|s)} - 1$, positive if $\pi(a|s) > \hat{\beta}(a|s)$ and otherwise negative, imposes a positive penalty on the $Q$-function when $\pi(a|s) > \hat{\beta}(a|s)$, and otherwise, it increases the $Q$-function since the penalty is negative, as the Q-penalization method considered in CQL [21].

To elaborate further on our proposed penalty, Fig. 2(a) depicts the log-probability of $\hat{\beta}$ and the thresholds $\tau$ used for penalty adaptation, with $N$ representing the number of data points. In Fig. 2(a), if the log-probability $\log\hat{\beta}$ of an action $a \in \mathcal{A}$ exceeds the threshold $\tau$, this indicates that the action $a$ is sufficiently represented in the dataset. Thus, we reduce the penalty for such actions. Furthermore, as shown in Fig. 2(a), when the number of actions increase from $N_1$ to $N_2$, the threshold for determining "enough data" decreases from $\tau_1$ to $\tau_2$, even if the data distribution remains unchanged.

Furthermore, to explain the role of the threshold $\tau$ in the proposed penalty $\mathcal{P}_\tau$, we consider two thresholds, $\tau_1$ and $\tau_2$. In Fig. 2(b), which illustrates the proposed penalty adaptation factor $f_{\tau_1}^{\pi,\hat{\beta}}$ and $f_{\tau_2}^{\pi,\hat{\beta}}$ for thresholds $\tau_1$ and $\tau_2$, $x_{\tau_1}^{\hat{\beta}}$ is larger than $x_{\tau_2}^{\hat{\beta}}$ because $\tau_1 > \tau_2$. As a result, in the case of $\tau_1$, $\mathcal{P}_{\tau_1}$ only reduces the penalty for $\pi_3$. In other words, $f_{\tau_1}^{\pi_1,\hat{\beta}} = f_{\tau_1}^{\pi_2,\hat{\beta}} = 1$, and $f_{\tau_1}^{\pi_3,\hat{\beta}} < 1$. On the other hand, as the number of data samples increases from $N_1$ to $N_2$, more actions generated by the behavior policy $\beta$ will be stored in the dataset, so policy actions are more likely to be in the dataset. In this case, the threshold should be lowered from $\tau_1$ to $\tau_2$. As a result, $\hat{\beta}$ exceeds the threshold $\tau_2$ in the support of all policies $\pi_i$, and $\mathcal{P}_{\tau_2}$ reduces the penalty in the support of all policies $\pi_i$, i.e., $f_{\tau_2}^{\pi_3,\hat{\beta}} < f_{\tau_2}^{\pi_1,\hat{\beta}} < f_{\tau_2}^{\pi_2,\hat{\beta}} < 1$. Thus, even without knowing the exact number of data samples, the proposed penalty $\mathcal{P}_\tau$ allows adjusting the penalty level appropriately according to the given number of data samples based on the threshold $\tau$.

Now, we propose exclusively penalized Q-learning (EPQ), a novel offline RL method that minimizes the Bellman error while imposing the proposed exclusive penalty $\mathcal{P}_\tau$ on the $Q$-function as follows:

$$\min_Q \ \mathbb{E}_{s,a,s'\sim D}\left[(Q(s,a) - \{\mathcal{B}^\pi Q(s,a) - \alpha\mathcal{P}_\tau\})^2\right]. \tag{3}$$

Then, we can prove that the final $Q$-function of EPQ underestimates the true value function $Q^\pi$ in offline RL if $\alpha$ is sufficiently large, as stated in the following theorem. This indicates that the proposed EPQ can successfully reduce overestimation bias in offline RL, while simultaneously alleviating unnecessary bias based on the proposed penalty $\mathcal{P}_\tau$.

**Theorem 3.1.** *We denote the $Q$-function converged from the $Q$-update of EPQ using the proposed penalty $\mathcal{P}_\tau$ in (3) by $\hat{Q}^\pi$. Then, the expected value of $\hat{Q}^\pi$ underestimates the expected true policy value, i.e., $\mathbb{E}_{a\sim\pi}[\hat{Q}^\pi(s,a)] \leq \mathbb{E}_{a\sim\pi}[Q^\pi(s,a)], \forall s \in D$, with high probability $1 - \delta$ for some $\delta \in (0, 1)$, if the penalizing factor $\alpha$ is sufficiently large. Furthermore, the proposed penalty reduces the average penalty for policy actions compared to the average penalty of CQL.*

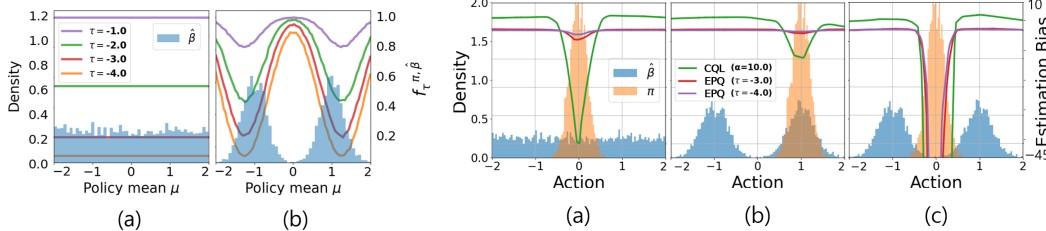

Figure 3: Histogram of $\hat{\beta}$ (left axis), and the corresponding $f_\tau^{\pi,\hat{\beta}}(s)$ with various $\tau$ (right axis) for two cases: (a) $\beta = \text{Unif}(-2,2)$ (b) $\beta = \frac{1}{2}N(-1,0.3) + \frac{1}{2}N(1,0.3)$

Figure 4: Histograms of $\pi$ and $\hat{\beta}$ (left axis), and the estimation bias of CQL and EPQ with various $\tau$ (right axis) for three cases: (a) $\beta = \text{Unif}(-2,2)$ and $\pi = N(0,0.2)$ (b) $\beta = \frac{1}{2}N(-1,0.3) + \frac{1}{2}N(1,0.3)$ and $\pi = N(1,0.2)$ (c) $\beta = \frac{1}{2}N(-1,0.3) + \frac{1}{2}N(1,0.3)$ and $\pi = N(0,0.2)$.

**Proof)** Proof of Theorem 3.1 is provided in Appendix A.

In order to demonstrate the $Q$-function convergence behavior of the proposed EPQ in more detail, we revisit the previous Pendulum task in Fig. 1. Fig. 3 shows the histogram of $\hat{\beta}$ and the penalty adaptation factor $f_\tau^{\pi,\hat{\beta}}(s)$ for Gaussian policy $\pi = N(\mu, 0.2)$, where $\mu$ varies from $-2$ to $2$, with varying $\beta$. In Fig. 3(a), $f_\tau^{\pi,\hat{\beta}}(s)$ should be less than 1 for any policy mean $\mu$ since all policy actions are sufficient in the dataset. In 3(b), $f_\tau^{\pi,\hat{\beta}}(s)$ is less than 1 only if the $\hat{\beta}$ probability near the policy mean $\mu$ is high, and otherwise, $f_\tau^{\pi,\hat{\beta}}(s)$ is 1, which indicates the lack of policy action in the dataset. Thus, the result shows that $f_\tau^{\pi,\hat{\beta}}(s)$ reflects our motivation in Section 3.1 well. Moreover, Fig. 4 compares the estimation bias curves of CQL and EPQ with $\alpha = 10$ in the scenarios presented in Fig. 1. CQL exhibits unnecessary bias for situations in Fig. 4(a) and Fig. 4(b) where no penalty is needed, as discussed in Section 3.1. Conversely, our proposed method effectively reduces estimation bias in these cases while appropriately maintaining the penalty for the scenario in Fig. 4(c) where penalization is required. This experiment demonstrates the effectiveness of our proposed approach, and the subsequent numerical results in Section 4 will numerically show that our method significantly reduces estimation bias in offline learning, resulting in improved performance.

### 3.3 Prioritized Dataset

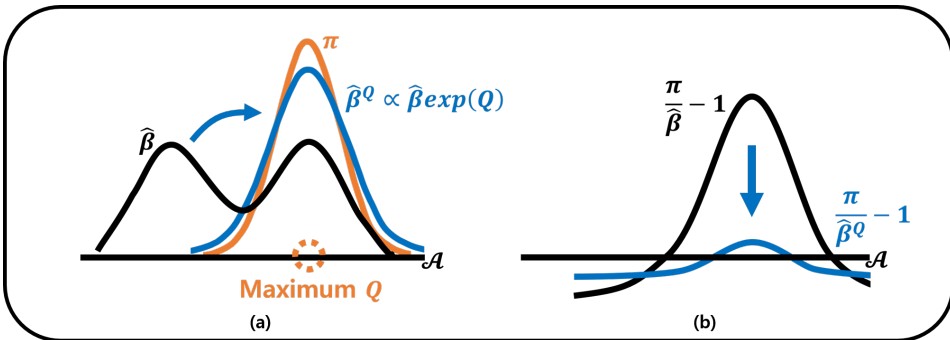

Figure 5: An illustration of the prioritized dataset. As the policy focuses on actions with maximum $Q$-values, the difference between $\hat{\beta}$ and $\pi$ becomes substantial, inducing large penalty: (a) The change of data distribution from $\hat{\beta}$ (w/o PD) to $\hat{\beta}^Q$ (with PD) (b) The corresponding penalty graphs for $\hat{\beta}$ (w/o PD) and $\hat{\beta}^Q$ (with PD).

In Section 3.2, EPQ effectively controls the penalty in the scenarios depicted in Fig. 4. However, in cases where the policy is highly concentrated on one side, as shown in Fig. 4, the estimation bias may not be completely eliminated due to the latter penalty term $\frac{\pi}{\hat{\beta}} - 1$ in $\mathcal{P}_\tau$, as $\pi$ significantly exceeds $\hat{\beta}$. This situation, detailed in Fig. 5, arises when there is a substantial difference in the $Q$-function values among data actions. As the policy is updated to maximize the $Q$-function, the policy shifts towards the data action with a larger $Q$, resulting in a more significant penalty for CQL. To further

alleviate the penalty to reduce unnecessary bias in this situation, instead of applying a penalty based on $\hat{\beta}$, we introduce a penalty based on the prioritized dataset (PD) $\hat{\beta}^Q \propto \hat{\beta} \exp(Q)$. As shown in Fig. 5(a), which illustrates the difference between the original data distribution $\hat{\beta}$ and the modified data distribution $\hat{\beta}^Q$ after applying PD, $\beta^Q$ prioritizes data actions with higher $Q$-values within the support of $\hat{\beta}$. According to Fig. 5(a), when the policy $\pi$ focuses on specific actions, the penalty $\frac{\pi}{\hat{\beta}} - 1$ increases significantly, as depicted in Fig. 5(b). In contrast, by applying PD, $\hat{\beta}$ is adjusted to approach $\hat{\beta}^Q \propto \beta \exp(Q)$, aligning the data distribution more closely with the policy $\pi$. Consequently, we anticipate that the penalty will be significantly mitigated, as the difference between $\pi$ and $\hat{\beta}^Q$ is much smaller than the difference between $\pi$ and $\hat{\beta}$. Following this intuition, we modify our penalty using PD as $\mathcal{P}_{\tau,\,PD} := f_\tau^{\pi,\hat{\beta}}(s) \cdot \left( \frac{\pi(a|s)}{\hat{\beta}^Q(a|s)} - 1 \right)$. It is important to note that the penalty adaptation factor $f_\tau^{\pi,\hat{\beta}}(s)$ remains unchanged since we use all data samples in the dataset for $Q$ updates. Additionally, we consider the prioritized dataset for the Bellman update to focus more on data actions with higher $Q$-function values for better performance as considered in [25]. Then, we can derive the final $Q$-loss function of EPQ with PD as

$$L(Q) = \frac{1}{2}\mathbb{E}_{s,s'\sim D, a\sim\hat{\beta}^Q}\left[ (Q - \{\mathcal{B}^\pi Q - \alpha\mathcal{P}_{\tau,\,PD}\})^2 \right] \tag{4}$$

$$= \mathbb{E}_{s,s'\sim D, a\sim\hat{\beta}, a'\sim\pi}\left[ w_{s,a}^Q \cdot \left\{ \frac{1}{2}\left(Q(s,a) - \mathcal{B}^\pi Q(s,a)\right)^2 + \alpha f_\tau^{\pi,\hat{\beta}}(s)(Q(s,a') - Q(s,a)) \right\} \right] + C,$$

where $w_{s,a}^Q = \frac{\hat{\beta}^Q(a|s)}{\hat{\beta}(a|s)} = \frac{\exp(Q(s,a))}{\mathbb{E}_{a'\sim\hat{\beta}(\cdot|s)}[\exp(Q(s,a'))]}$ is the importance sampling (IS) weight, $C$ is the remaining constant term, and the detailed derivation of (4) is provided in Appendix B.1. The ablation study in Section 4.3 will show that EPQ performs better when prioritized dataset $\hat{\beta}^Q$ is considered.

### 3.4 Practical Implementation and Algorithm

Now, we propose the implementation of EPQ based on the value loss function (4). Basically, our implementation follows the setup of CQL [21]. For policy, we utilize the Gaussian policy with a $\text{Tanh}(\cdot)$ layer proposed by Haarnoja et al. [4] and update the policy to maximize the $Q$-function with its entropy. Then, the policy loss function is given by

$$L(\pi) = \mathbb{E}_{s\sim D,\ a\sim\pi}[-Q(s,a) + \log\pi(a|s)]. \tag{5}$$

Based on the $Q$-update in (4) and the policy loss function (5), we summarize the algorithm of EPQ as Algorithm 1. More detailed implementation, including the calculation method of the IS weight $w_{s,a}^Q$ and redefined loss functions for the parameterized $Q$ and $\pi$, is provided in Appendix B.2.

---

**Algorithm 1** Exclusively Penalized Q-learning

---

**Require:** Offline dataset $D$
1: Train the behavior policy $\hat{\beta}$ based on behavior cloning (BC)
2: Initialize $Q$ and $\pi$
3: **for** gradient step $k = 0, 1, 2, 3, \ldots$ **do**
4:   Sample batch transitions $\{(s, a, r, s')\}$ from $D$.
5:   Calculate the penalty adaptation factor $f_\tau^{\pi,\hat{\beta}}(s)$ and IS weight $w_{s,a}^Q$
6:   Compute losses $L(Q)$ in Equation (4) and $L(\pi)$ in Equation (5)
7:   Update the policy $\pi$ to minimize $L(\pi)$
8:   Update the $Q$-function $Q$ to minimize $L(Q)$
9: **end for**

---

## 4 Experiments

In this section, we evaluate our proposed EPQ against other state-of-the-art offline RL algorithms using the D4RL benchmark [23], commonly used in the offline RL domain. Among various D4RL tasks, we mainly consider Mujoco locomotion tasks, Adroit manipulation tasks, and AntMaze navigation tasks, with scores normalized from 0 to 100, where 0 represents random performance and 100 represents expert performance.

Table 1: Performance comparison: Normalized average return results

| Task name | BC | 10% BC | TD3+BC | CQL (paper) | CQL (reprod.) | Onestep | IQL | MCQ | MISA | EPQ |
|---|---|---|---|---|---|---|---|---|---|---|
| halfcheetah-random | 2.3 | 2.2 | 12.7 | **35.4** | 20.8 | 6.9 | 12.9 | 28.5 | 2.5 | 33.0±2.4 |
| hopper-random | 4.1 | 4.7 | 22.5 | 10.8 | 9.7 | 7.9 | 9.6 | 31.8 | 9.9 | **32.1±0.3** |
| walker2d-random | 1.7 | 2.3 | 7.2 | 7.0 | 7.1 | 6.2 | 6.9 | 17.0 | 9.0 | **23.0±0.7** |
| halfcheetah-medium | 42.6 | 42.5 | 48.3 | 44.4 | 44.0 | 48.4 | 47.4 | 64.3 | 47.4 | **67.3±0.5** |
| hopper-medium | 52.9 | 56.9 | 59.3 | 86.6 | 58.5 | 59.6 | 66.3 | 78.4 | 67.1 | **101.3±0.2** |
| walker2d-medium | 75.3 | 75.0 | 83.7 | 74.5 | 72.5 | 81.8 | 78.3 | **91.0** | 84.1 | 87.8±2.1 |
| halfcheetah-medium-expert | 55.2 | 92.9 | 90.7 | 62.4 | 91.6 | 93.4 | 86.7 | 87.5 | 94.7 | **95.7±0.3** |
| hopper-medium-expert | 52.5 | 110.9 | 98.0 | 111.0 | 105.4 | 103.3 | 91.5 | **111.2** | 109.8 | 108.8±5.2 |
| walker2d-medium-expert | 107.5 | 109.0 | 110.1 | 98.7 | 108.8 | 113.0 | 109.6 | **114.2** | 109.4 | 112.0±0.6 |
| halfcheetah-expert | 92.9 | 91.9 | 98.6 | 104.8 | 96.3 | 92.3 | 95.4 | 96.2 | 95.9 | **107.2±0.2** |
| hopper-expert | 111.2 | 109.6 | 111.7 | 109.9 | 110.8 | 112.3 | **112.4** | 111.4 | 111.9 | **112.4±0.5** |
| walker2d-expert | 108.5 | 109.1 | 110.3 | **121.6** | 110.0 | 111.0 | 110.1 | 107.2 | 109.3 | 109.8±1.0 |
| halfcheetah-medium-replay | 36.6 | 40.6 | 44.6 | 46.2 | 45.5 | 38.1 | 44.2 | 56.8 | 45.6 | **62.0±1.6** |
| hopper-medium-replay | 18.1 | 75.9 | 60.9 | 48.6 | 95.0 | 97.5 | 94.7 | **101.6** | 98.6 | 97.8±1.0 |
| walker2d-medium-replay | 26.0 | 62.5 | 81.8 | 32.6 | 77.2 | 49.5 | 73.9 | **91.3** | 86.2 | 85.3±1.0 |
| halfcheetah-full-replay | 62.4 | 68.7 | 75.9 | - | 76.9 | 80.0 | 73.3 | 82.3 | 74.8 | **85.3±0.7** |
| hopper-full-replay | 34.3 | 92.8 | 81.5 | - | 101.0 | 107.8 | 107.2 | **108.5** | 103.5 | **108.5±0.6** |
| walker2d-full-replay | 45.0 | 89.4 | 95.2 | - | 93.4 | 102.0 | 98.1 | 95.7 | 94.8 | **107.4±0.6** |
| **Mujoco Tasks Total** | 929.1 | 1236.9 | 1293.0 | - | 1325.8 | 1311.0 | 1318.5 | 1474.9 | 1354.5 | **1536.7** |
| pen-human | 63.9 | -2.0 | 64.8 | 55.8 | 37.5 | 71.8 | 71.5 | 68.5 | **88.1** | 83.9±6.8 |
| door-human | 2.0 | 0.0 | 0.0 | 9.1 | 9.9 | 5.4 | 4.3 | 2.3 | 5.2 | **13.2±2.4** |
| hammer-human | 1.2 | 0.0 | 1.8 | 2.1 | 4.4 | 1.2 | 1.4 | 0.3 | **8.1** | 3.9±5.0 |
| relocate-human | 0.1 | 0.0 | 0.1 | 0.4 | 0.2 | **1.9** | 0.1 | 0.1 | 0.1 | 0.3±0.2 |
| pen-cloned | 37.0 | 0.0 | 49 | 40.3 | 39.2 | 60.0 | 37.3 | 49.4 | 58.6 | **91.8±4.7** |
| door-cloned | 0.0 | 0.0 | 0.0 | 3.5 | 0.4 | 0.4 | 1.6 | 1.3 | 0.5 | **5.8±2.8** |
| hammer-cloned | 0.6 | 0.0 | 0.2 | 5.7 | 2.1 | 2.1 | 2.1 | 1.4 | 2.2 | **22.8±15.3** |
| relocate-cloned | -0.3 | 0.0 | -0.2 | -0.1 | -0.1 | -0.1 | -0.2 | 0.0 | -0.1 | **0.1±0.1** |
| **Adroit Tasks Total** | 104.5 | -2 | 115.7 | 116.8 | 93.6 | 142.7 | 118.1 | 123.3 | 162.7 | **221.8** |
| umaze | 54.6 | 62.8 | 78.6 | 74.0 | 80.4 | 72.5 | 87.5 | 98.3 | 92.3 | **99.4±1.0** |
| umaze-diverse | 45.6 | 50.2 | 71.4 | 84.0 | 56.3 | 75.0 | 62.2 | 80.0 | **89.1** | 78.3±5.0 |
| medium-play | 0.0 | 5.4 | 10.6 | 61.2 | 67.5 | 5.0 | 71.2 | 52.5 | 63.0 | **85.0±11.2** |
| medium-diverse | 0.0 | 9.8 | 3.0 | 53.7 | 62.5 | 5.0 | 70.0 | 37.5 | 62.8 | **86.7±18.9** |
| large-play | 0.0 | 0.0 | 0.2 | 15.8 | 35.0 | 2.5 | 39.6 | 2.5 | 17.5 | **40.0±8.2** |
| large-diverse | 0.0 | 6.0 | 0.0 | 14.9 | 13.3 | 2.5 | **47.5** | 7.5 | 23.4 | 36.7±4.7 |
| **AntMaze Tasks Total** | 100.2 | 134.2 | 163.8 | 303.6 | 315.0 | 162.5 | 378.0 | 278.3 | 348.1 | **426.1** |

**Mujoco Locomotion Tasks:** The D4RL dataset comprises offline datasets obtained from Mujoco tasks [26] like HalfCheetah, Hopper, and Walker2d. Each task has 'random', 'medium', and 'expert' datasets, obtained by a random policy, the medium policy with performance of 50 to 100 points, and the expert policy with performance of 100 points, respectively. Additionally, there are 'medium-expert' dataset that contains both 'medium' and 'expert' data, 'medium-replay' and 'full-replay' datasets that contain the buffers generated while the medium and expert policies are trained, respectively.

**Adroit Manipulation Tasks:** Adroit provides four complex manipulation tasks: Pen, Hammer, Door, and Relocate, utilizing motion-captured human data with associated rewards. Each task has two datasets: 'human' dataset derived from human motion-capture data, and 'cloned' dataset comprising samples from both the cloned behavior policy using BC and the original motion-capture data.

**AntMaze Navigation Tasks:** AntMaze is composed of six navigation tasks including 'umaze', 'umaze-diverse', 'medium-play', 'medium-diverse', 'large-play', and 'large-diverse' where robot ant agent is trained to reach a goal within the maze. While 'play' dataset is acquired under a fixed set of goal locations and a fixed set of starting locations, the 'diverse' dataset is acquired under a random goal locations and random starting locations setting.

## 4.1 Performance Comparisons

We compare our algorithm with various constraint-based offline RL methods, including CQL baselines [21] on which our algorithm is based on. For other baseline methods, we consider behavior cloning (BC) and 10% BC, where the latter only utilizes only the top 10% of demonstrations with high returns, TD3+BC [27] that simply combines BC with TD3 [3], Onestep RL [28] that performs a single policy iteration based on the dataset, implicit $Q$-learning (IQL) [29] that seeks the optimal value function for the dataset through expectile regression, mildly conservative $Q$-learning (MCQ) [30] that reduces overestimation by using pseudo $Q$ values for out-of-distribution actions, and MISA [31] that considers the policy constraint based on mutual information. To assess baseline algorithm performance, we utilize results directly from the original papers for CQL (paper) [21] and MCQ

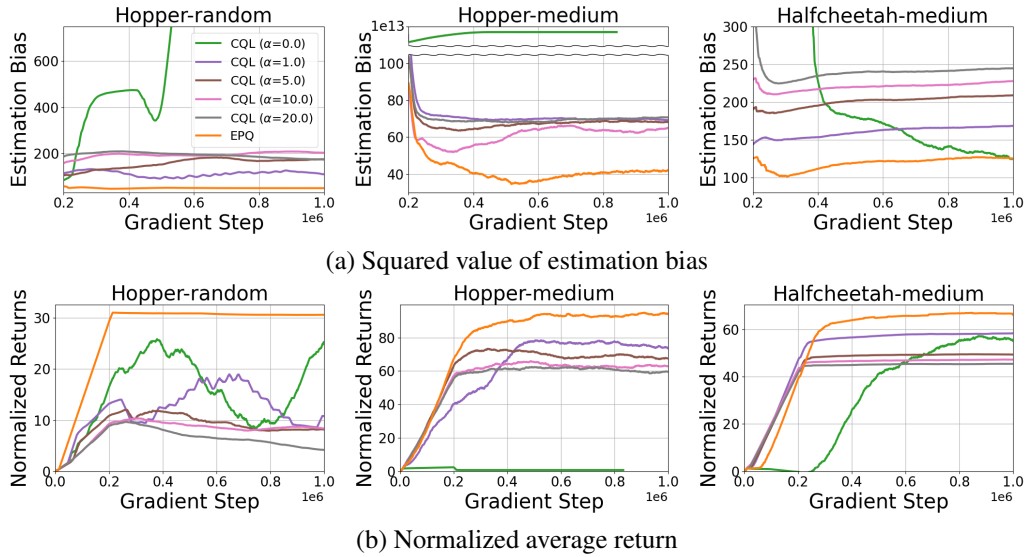

(a) Squared value of estimation bias

(b) Normalized average return

Figure 6: Analysis of proposed method

[30], as well as reported results from other baseline algorithms according to Ma et al. [31]. For CQL, reproducing its performance is challenging, so we also include reproduced CQL performance labeled as CQL (reprod.) from Ma et al. [31]. Any missing experimental results have been filled in by re-implementing each baseline algorithm. For our algorithm, we explored various penalty control thresholds $\tau \in \{c \cdot \rho, \ c \in [0, 10]\}$, where $\rho$ represents the log-density of $\text{Unif}(\mathcal{A})$. For Mujoco tasks, the EPQ penalizing constant is fixed at $\alpha = 20.0$, and for Adroit tasks, we consider either $\alpha = 5.0$ or $\alpha = 20.0$. To ensure robustness, we run our algorithm with four different seeds for each task. Table 1 displays the average normalized returns and corresponding standard deviations for compared algorithms. The performance of EPQ is based on the best hyperparameter setup, with additional results presented in the ablation study in Section 4.3. Further details on the hyperparameter setup are provided in Appendix C.

The results in Table 1 shows that our algorithm significantly outperforms the other constraint-based offline RL algorithms in all considered tasks. In particular, in challenging tasks such as Adroit tasks and AntMaze tasks, where rewards are sparse or intermittent, EPQ demonstrates remarkable performance improvements compared to recent offline RL methods. This is because EPQ can impose appropriate penalty on each state, even if the policy and behavior policy varies depending on the timestep as demonstrated in Section 3.2. Also, we observe that our proposed algorithm shows a large increase in performance in the 'Hopper-random', 'Hopper-medium', and 'Halfcheetah-medium' environments compared to CQL, so we will further analyze the causes of the performance increase in these tasks in the following section. For adroit tasks, the performance of CQL (reprod.) is too low compared to CQL (paper), so we provide the enhanced version of CQL in Appendix E, but the result in Appendix E shows that EPQ still performs better than the enhanced version of CQL.

## 4.2 The Analysis of Estimation Bias

In Section 4.1, EPQ outperforms CQL baselines significantly across various D4RL tasks based on our proposed penalty in Section 3. To analyze the impact of our overestimation reduction method on performance enhancement, we compare the estimation bias for EPQ and CQL baselines with various penalizing constants $\alpha \in \{0, 1, 5, 10, 20\}$ on 'Hopper-random', 'Hopper-medium', and 'Halfcheetah-medium' tasks. In Fig. 6(a), we depict the squared value of estimation bias, obtained from the difference between the $Q$-value and the empirical average return for sample trajectories generated by the policy, to show both overestimation bias and underestimation bias. In the experiment shown in Fig. 6(a), the estimation bias in CQL with $\alpha = 0$ became excessively large, causing the gradients to explode and resulting in forced termination of the training. Fig. 6(b) illustrates the corresponding normalized average returns, emphasizing learning progress after 200k gradient steps.

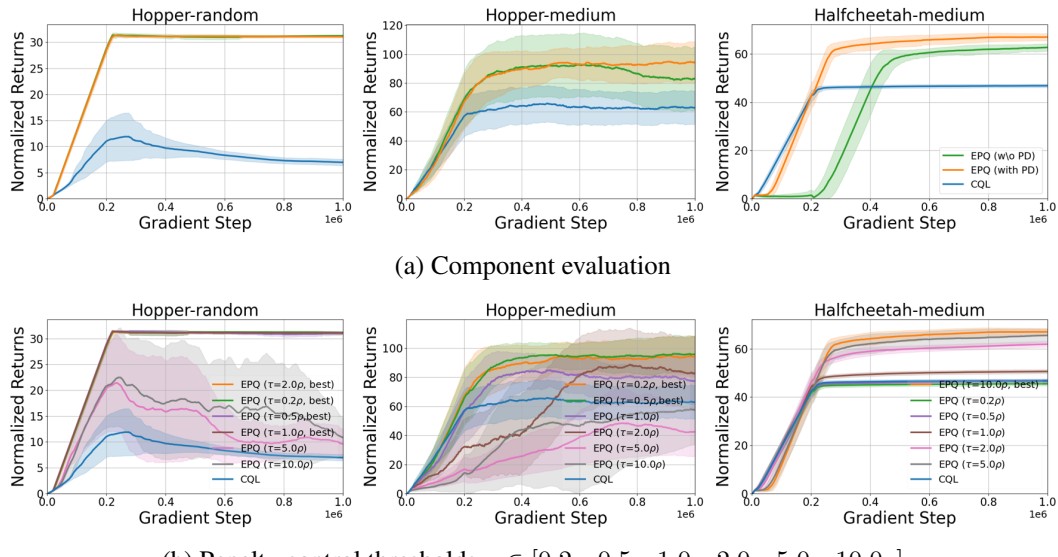

(a) Component evaluation

(b) Penalty control thresholds $\tau \in [0.2\rho, 0.5\rho, 1.0\rho, 2.0\rho, 5.0\rho, 10.0\rho]$

Figure 7: Additional ablation studies on the Hopper-random, Hopper-medium, and Halfcheetah-medium tasks are presented. The best hyperparameter in the paper is denoted by the orange curve.

In Fig. 6(a), we observe an increase in estimation bias for CQL as the penalizing constant $\alpha$ rises, attributed to unnecessary bias highlighted in Fig. 1. Reducing $\alpha$ to nearly 0 in CQL, however, fails to effectively mitigate overestimation error, leading to a divergence of the $Q$-function in tasks such as 'Hopper-random' and 'Hopper-medium', as shown in Fig. 1. Conversely, EPQ demonstrates superior reduction of estimation bias in the $Q$-function compared to CQL baselines for all tasks in Fig. 6(a), indicating its capability to mitigate both overestimation and underestimation bias based on the proposed penalty. As a result, Fig. 6(b) shows that EPQ significantly outperforms all CQL variants on 'Hopper-random', 'Hopper-medium', and 'Halfcheetah-medium' tasks.

### 4.3 Ablation Study

To understand the impact of EPQ's components and hyperparameters, we conduct ablation studies to evaluate each component and the penalty control threshold $\tau$ on the 'Hopper-random', 'Hopper-medium', and 'HalfCheetah-medium' tasks where our proposed method showed a significant performance improvement compared to the baseline CQL.

**Component Evaluation:** In Section 3, we introduced two variants of the EPQ algorithm: EPQ (w/o PD), which does not incorporate a prioritized dataset as in equation (3), and EPQ (with PD), which leverages a prioritized dataset based on $\hat{\beta}^Q$ as in equation (4). In Fig. 7(a), we compare the performance of EPQ (w/o PD), EPQ (with PD), and the CQL baseline to analyze the impact of each component. EPQ (w/o PD) still outperforms CQL, demonstrating that the proposed penalty $\mathcal{P}_\tau$ in Section 3.2 enhances performance by efficiently reducing overestimation without introducing unnecessary estimation bias, as discussed in Section 3.2. Additionally, Fig. 7(a) shows that EPQ (with PD) outperforms EPQ (w/o PD) significantly in the HalfCheetah-medium task, indicating that the proposed prioritized dataset contributes to improved performance, as anticipated in Section 3.3.

**Penalty Control Threshold $\tau$:** As discussed in Section 3.2, EPQ can dynamically control the penalty amount based on the penalty control threshold $\tau$, as illustrated in Fig. 2, even in the absence of knowledge about the exact number of data samples. Fig. 7(b) demonstrates the performance of EPQ with various penalty control thresholds $\tau \in [0.2\rho, 0.5\rho, 1.0\rho, 2.0\rho, 5.0\rho, 10.0\rho]$, where $\rho$ represents the log-density of $\text{Unif}(\mathcal{A})$. Note that $\rho$ is negative, so $\tau = 10.0\rho$ is the lowest threshold while $\tau = 0.2\rho$ is the highest. The results indicate that in tasks like Hopper-medium, where a variety of actions are not sufficiently sampled, a higher threshold performs better. Conversely, in tasks like Hopper-random, where a broad range of actions is sampled, a lower threshold is more effective. An exception is the HalfCheetah-medium task, which, despite having fewer action variations, visits a diverse range of states. This may result in lower overestimation errors for OOD

actions, benefiting from a lower threshold. Furthermore, the performance on the considered tasks appears to be surprisingly less sensitive to changes in $\tau$. We initially expect that performance might be sensitive to $\tau$ since it reflects the fixed number of data samples, but the results indicate that the performance is not significantly affected by variations in $\tau$. Moreover, EPQ algorithms with different $\tau$ consistently outperform the CQL baseline, highlighting the superiority of the proposed method.

## 5 Related Works

### 5.1 Constraint-based Offline RL

In order to reduce the overestimation in offline learning, several constraint-based offline RL methods have been studied. Fujimoto et al. [17] propose a batch-constrained policy to minimize the extrapolation error, and Kumar et al. [19], Wu et al. [20] limits the distribution based on the distance of the distribution, rather than directly constraining the policy. Fujimoto and Gu [27] restricts the policy actions to batch data based on the online algorithm TD3 [3]. Furthermore, Kumar et al. [21], Yu et al. [32] aims to minimize the probability of out-of-distribution actions using the lower bound of the true value. By predicting a more optimistic cost for tuples within the batch data, Xu et al. [22] provides stable training for offline-based safe RL tasks. On the other hand, Ma et al. [31] utilizes mutual information to constrain the policy.

### 5.2 Offline Learning based on Data Optimality

In offline learning setup, the optimality of the dataset greatly impacts the performance [25]. Simply using $n$-% BC, or applying weighted experiences, [33, 34] which utilize only a portion of the data based on the evaluation results of the given data, fails to exploit the distribution of the data. Based on Haarnoja et al. [35], Reddy et al. [36], Garg et al. [37] uses the Boltzmann distribution for offline learning, training the policy to follow actions with higher value in the imitation learning domain [38, 39]. Kostrikov et al. [29] and Xiao et al. [40] argue that the optimality of data can be improved by using expectile regression and in-sample SoftMax, respectively. Additionally, methods that learn the value function from the return of the data in a supervised manner have been proposed [41, 28, 42, 43].

### 5.3 Value Function Shaping

In offline RL, imposing constraints on the policy can decrease the performance, thus Kumar et al. [21], Lyu et al. [44] impose penalties on out-of-distribution actions by structuring the learned value function as a lower bound to the actual values. Additionally, Fakoor et al. [45] addresses the issue by imposing a policy constraint based on divergence and suppressing overly optimistic estimations on the value function, thereby preventing excessive expansion of the value function. Moreover, Wu et al. [46] predicts the instability of actions through the variance of the value function, imposing penalties on the out-of-distribution actions, while Lyu et al. [30] replaces the $Q$ values for out-of-distribution actions with pseudo $Q$-values and Agarwal et al. [47], An et al. [48], Bai et al. [49], Lee et al. [50] mitigates the instability of learning the value function by applying ensemble techniques. In addition, Ghosh et al. [51] interprets the changes in MDP from a Bayesian perspective through the value function, thereby conducting adaptive policy learning.

## 6 Conclusion

To mitigate overestimation error in offline RL, this paper focuses on exclusive penalty control, which selectivelys gives the penalty for states where policy actions are insufficient in the dataset. Furthermore, we propose a prioritized dataset to enhance the efficiency of reducing unnecessary bias due to the penalty. As a result, our proposed method, EPQ, successfully reduces the overestimation error arising from distributional shift, while avoiding underestimation error due to the penalty. This significantly reduces estimation bias in offline learning, resulting in substantial performance improvements across various D4RL tasks.

## Acknowledgements

This work was supported in part by Institute of Information & Communications Technology Planning & Evaluation (IITP) grant funded by the Korea government (MSIT) (No.2022-0-00469, Development of Core Technologies for Task-oriented Reinforcement Learning for Commercialization of Autonomous Drones) and in part by IITP grant funded by the Korea government (MSIT) (No. RS-2022-00156361, Innovative Human Resource Development for Local Intellectualization(UNIST)) and in part by Artificial Intelligence Graduate School support (UNIST), IITP grant funded by the Korea government (MSIT) (No.2020-0-01336).

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

# A Proof

**Theorem 3.1** *We denote the Q-function converged from the Q-update of EPQ using the proposed penalty $\mathcal{P}_\tau$ in (3) by $\hat{Q}^\pi$. Then, the expected value of $\hat{Q}^\pi$ underestimates the expected true policy value, i.e., $\mathbb{E}_{a\sim\pi}[\hat{Q}^\pi(s,a)] \leq \mathbb{E}_{a\sim\pi}[Q^\pi(s,a)], \forall s \in D$, with high probability $1-\delta$ for some $\delta \in (0,1)$, if the penalizing factor $\alpha$ is sufficiently large. Furthermore, the proposed penalty reduces the average penalty for policy actions compared to the average penalty of CQL.*

## A.1 Proof of Theorem 3.1

Proof of Theorem 3.1 basically follows the proof of Theorem 3.2 in Kumar et al. [21] since $\mathcal{P}_\tau$ multiplies the penalty control factor $f_\tau^{\pi,\hat{\beta}}(s)$ to the penalty of CQL. At each $k$-th iteration, $Q$-function is updated by equation (4), then

$$Q_{k+1}(s,a) \leftarrow \hat{\mathcal{B}}^\pi Q_k(s,a) - \alpha\mathcal{P}_\tau, \ \forall s,a, \tag{A.1}$$

where $\hat{\mathcal{B}}^\pi$ is the estimation of the true Bellman operator $\mathcal{B}^\pi$ based on data samples. It is known that the error between the estimated Bellman operator $\hat{\mathcal{B}}^\pi$ and the true Bellman operator is bounded with high probability of $1-\delta$ for some $\delta \in (0,1)$ as $|(\mathcal{B}^\pi Q)(s,a) - (\hat{\mathcal{B}}^\pi Q)(s,a)| \leq \xi^\delta(s,a), \ \forall s,a$, where $\xi^\delta$ is a positive constant related to the given dataset $D$, the discount factor $\gamma$, and the transition probability $P$ [21]. Then, with high probability $1-\delta$,

$$Q_{k+1}(s,a) \leftarrow \mathcal{B}^\pi Q_k(s,a) - \alpha\mathcal{P}_\tau + \xi^\delta(s,a), \ \forall s,a, \tag{A.2}$$

Now, with the state value function $V(s) := \mathbb{E}_{a\sim\pi(\cdot|s)}[Q(s,a)]$

$$V_{k+1}(s) = \mathbb{E}_{a\sim\pi(\cdot|s)}[Q_k(s,a)] = \mathcal{B}^\pi V_k - \alpha\mathbb{E}_{a\sim\pi}[\mathcal{P}_\tau] + \xi^\delta(s,a)$$

$$= \mathcal{B}^\pi V_k(s) - \alpha\mathbb{E}_{a\sim\pi}\left[f_\tau^{\pi,\hat{\beta}}(s) \cdot \left(\frac{\pi(a|s)}{\hat{\beta}(a|s)} - 1\right) + \mathbb{E}_{a\sim\pi}[\xi^\delta(s,a)]\right]$$

$$= \mathcal{B}^\pi V_k(s) - \alpha\Delta_{EPQ}^\pi(s) + \mathbb{E}_{a\sim\pi}[\xi^\delta(s,a)] \tag{A.3}$$

Upon repeated iteration, $V_{k+1}$ converges to $V_\infty(s) = V^\pi(s) + (I - \gamma P^\pi)^{-1} \cdot \{-\alpha\Delta_{EPQ}^\pi(s) + \mathbb{E}_{a\sim\pi}[\xi^\delta(s,a)]\}$ based on the fixed point theorem, where $\Delta_{EPQ}^\pi(s) := \mathbb{E}_{a\sim\pi}[\mathcal{P}_\tau]$ is the average penalty for policy $\pi$, $I$ is the identity matrix, and $P^\pi$ is the state transition matrix where the policy $\pi$ is given. Here, we can show that the average penalty $\Delta_{EPQ}^\pi(s)$ is positive as follows:

$$\Delta_{EPQ}^\pi(s) = \mathbb{E}_{a\sim\pi}\left[f_\tau^{\pi,\hat{\beta}}(s) \cdot \left(\frac{\pi(a|s)}{\hat{\beta}(a|s)} - 1\right)\right]$$

$$= f_\tau^{\pi,\hat{\beta}}(s)\left[\sum_{a\in\mathcal{A}}\pi(a|s)\left(\frac{\pi(a|s)}{\hat{\beta}(a|s)} - 1\right) - \underbrace{\sum_{a\in\mathcal{A}}\hat{\beta}(a|s)\left(\frac{\pi(a|s)}{\hat{\beta}(a|s)} - 1\right)}_{=0}\right]$$

$$= f_\tau^{\pi,\hat{\beta}}(s) \cdot \sum_{a\in\mathcal{A}}\frac{(\pi(a|s) - \hat{\beta}(a|s))^2}{\hat{\beta}(a|s)} \geq 0, \tag{A.4}$$

where the equality in (A.4) satisfies when $\pi = \hat{\beta}$ or $f_\tau^{\pi,\hat{\beta}} = 0$. Given that $V_{k+1}$ converges to $V_\infty = V^\pi(s) + (I - \gamma P^\pi)^{-1} \cdot \{-\alpha\Delta_{EPQ}^\pi(s) + \mathbb{E}_{a\sim\pi}[\xi^\delta(s,a)]\}$, choosing the penalizing constant $\alpha$ that satisfies $\alpha \geq \max_{s,a\in D}[\xi^\delta(s,a)] \cdot \max_{s\in D}(\Delta_{EPQ}^\pi(s))^{-1}$ will satisfy,

$$-\alpha \cdot \Delta_{EPQ}^\pi(s) + \mathbb{E}_{a\sim\pi}[\xi^\delta(s,a)]$$

$$\leq -\max_{s,a\in D}[\xi^\delta(s,a)] \cdot \underbrace{\max_{s\in D}(\Delta_{EPQ}^\pi(s))^{-1} \cdot \Delta_{EPQ}^\pi(s)}_{\geq 1} + \mathbb{E}_{a\sim\pi}[\xi^\delta(s,a)]$$

$$\leq -\max_{s,a\in D}[\xi^\delta(s,a)] + \mathbb{E}_{a\sim\pi}[\xi^\delta(s,a)] \leq 0, \quad \forall s, \tag{A.5}$$

Since $I - \gamma P^\pi$ is non-singular $M$-matrix and the inverse of non-singular $M$-matrix is non-negative, i.e., all elements of $(I - \gamma P^\pi)^{-1}$ are non-negative, $V_\infty(s) = V^\pi(s) + (I - \gamma P^\pi)^{-1} \cdot \{-\alpha \Delta_{EPQ}^\pi(s) + \mathbb{E}_{a \sim \pi}[\xi^\delta(s, a)]\} \leq V^\pi(s)$, $\forall s$. Therefore, $V_\infty$ underestimates the true value function $V^\pi$ if the penalizing constant $\alpha$ satisfies $\alpha \geq \max_{s,a \in D}[\xi^\delta(s, a)] \cdot \max_{s \in D}(\Delta_{EPQ}^\pi(s))^{-1}$. In addition, according to [21], the average penalty of CQL for policy actions can be represented as $\Delta_{CQL}^\pi(s) = \mathbb{E}_{a \sim \pi}[\frac{\pi}{\hat{\beta}} - 1]$. Thus, $\Delta_{EPQ}^\pi(s) = f_\tau^{\pi,\hat{\beta}}(s) \Delta_{CQL}^\pi(s)$ and $f_\tau^{\pi,\hat{\beta}}(s) \leq 1$ from the definition in (2), so $0 \leq \Delta_{EPQ}^\pi(s) \leq \Delta_{CQL}^\pi(s)$. In addition, if $\pi = \hat{\beta}$, then $0 = \Delta_{EPQ}^{\hat{\beta}}(s) = \Delta_{CQL}^{\hat{\beta}}(s)$ from the equality condition in (A.4), which indicates that the average penalty for data actions is 0 for both EPQ and CQL. $\blacksquare$

# B  Implementation Details

In this section, we provide the implementation details of the proposed EPQ. First of all, we provide a detailed derivation of the final $Q$-loss function(4) of EPQ in Section B.1. Next, we introduce a practical implementation of EPQ to compute the loss functions for the parameterized policy and $Q$-function in Section B.2. In addition, to calculate loss functions in Section B.2, we provide the additional implementation details in Appendices B.3, B.4, and B.5. We conduct our experiments on a single server equipped with an Intel Xeon Gold 6336Y CPU and one NVIDIA RTX A5000 GPU, and we compare the running time of EPQ with other baseline algorithms in Section B.6. For additional hyperparameters in the practical implementation of EPQ, we provide detailed hyperparameter setup and additional ablation studies in Appendix C and Appendix D, respectively.

## B.1  Detailed Derivation of $Q$-Loss Function

In Section 3.3, the final $Q$-loss function with the proposed penalty $\mathcal{P}_{\tau,PD} = f_\tau^{\pi,\hat{\beta}}(\frac{\pi}{\hat{\beta}Q} - 1)$ is given by $L(Q) = \frac{1}{2}\mathbb{E}_{s,s'\sim D,a\sim\hat{\beta}Q}\left[(Q - \{\mathcal{B}^\pi Q - \alpha\mathcal{P}_{\tau,\,PD}\})^2\right]$. In this section, we provide a more detailed calculation of $L(Q)$ to obtain (4) as follows:

$$
\begin{aligned}
L(Q) &= \frac{1}{2}\mathbb{E}_{s,s'\sim D,a\sim\hat{\beta}Q}\left[(Q - \{\mathcal{B}^\pi Q - \alpha\mathcal{P}_{\tau,\,PD}\})^2\right] \\
&= \frac{1}{2}\mathbb{E}_{s,s'\sim D,a\sim\hat{\beta}Q}\left[(Q - \mathcal{B}^\pi Q)^2\right] + \alpha\mathbb{E}_{s,s'\sim D,a\sim\hat{\beta}Q}\left[\mathcal{P}_{\tau,\,PD}\cdot Q\right] + C \\
&= \frac{1}{2}\mathbb{E}_{s,s'\sim D,a\sim\hat{\beta}Q}\left[(Q - \mathcal{B}^\pi Q)^2\right] + \alpha\mathbb{E}_{s,s'\sim D,a\sim\hat{\beta}Q}\left[f_\tau^{\pi,\hat{\beta}}\left(\frac{\pi}{\hat{\beta}Q} - 1\right)Q\right] + C \\
&= \frac{1}{2}\mathbb{E}_{s,s'\sim D,a\sim\hat{\beta}Q}\left[(Q - \mathcal{B}^\pi Q)^2\right] + \alpha\mathbb{E}_{s,s'\sim D}\left[\int_{a\in\mathcal{A}}\hat{\beta}Q f_\tau^{\pi,\hat{\beta}}\left(\frac{\pi}{\hat{\beta}Q} - 1\right)Q da\right] + C \\
&= \frac{1}{2}\mathbb{E}_{s,s'\sim D,a\sim\hat{\beta}Q}\left[(Q - \mathcal{B}^\pi Q)^2\right] + \alpha\mathbb{E}_{s,s'\sim D}\left[\int_{a\in\mathcal{A}}f_\tau^{\pi,\hat{\beta}}\left(\pi - \hat{\beta}Q\right)Q da\right] + C \\
&= \frac{1}{2}\mathbb{E}_{s,s'\sim D,a\sim\hat{\beta}Q}\left[(Q - \mathcal{B}^\pi Q)^2\right] + \alpha\mathbb{E}_{s,s'\sim D}\left[\int_{a'\in\mathcal{A}}\pi f_\tau^{\pi,\hat{\beta}}Q da' - \int_{a\in\mathcal{A}}\hat{\beta}Q f_\tau^{\pi,\hat{\beta}}Q da\right] + C \\
&= \frac{1}{2}\mathbb{E}_{s,s'\sim D,a\sim\hat{\beta}Q}\left[(Q - \mathcal{B}^\pi Q)^2\right] + \alpha\mathbb{E}_{s,s'\sim D}\left[\mathbb{E}_{a'\sim\pi}\left[f_\tau^{\pi,\hat{\beta}}Q\right] - \mathbb{E}_{a\sim\hat{\beta}Q}\left[f_\tau^{\pi,\hat{\beta}}Q\right]\right] + C \\
&= \frac{1}{2}\mathbb{E}_{s,s'\sim D,a\sim\hat{\beta}Q}\left[(Q - \mathcal{B}^\pi Q)^2\right] + \alpha\mathbb{E}_{s,s'\sim D,a\sim\hat{\beta}Q}\left[\mathbb{E}_{a'\sim\pi}\left[f_\tau^{\pi,\hat{\beta}}Q\right] - f_\tau^{\pi,\hat{\beta}}Q\right] + C \\
&\underset{(*)}{=} \mathbb{E}_{s,s'\sim D,a\sim\hat{\beta}}\left[\frac{\hat{\beta}Q}{\hat{\beta}}\cdot\left\{\frac{1}{2}\left(Q - \mathcal{B}^\pi Q\right)^2 + \alpha f_\tau^{\pi,\hat{\beta}}\cdot\left(\mathbb{E}_{a'\sim\pi}\left[Q\right] - Q\right)\right\}\right] + C \\
&= \mathbb{E}_{s,s'\sim D,a\sim\hat{\beta},a'\sim\pi}\left[w_{s,a}^Q\cdot\left\{\frac{1}{2}\left(Q(s,a) - \mathcal{B}^\pi Q(s,a)\right)^2 + \alpha f_\tau^{\pi,\hat{\beta}}(s)(Q(s,a') - Q(s,a))\right\}\right] + C,
\end{aligned}
$$

where $C$ is the remaining constant term that can be ignored for the $Q$-update since $\mathcal{B}^\pi Q$ is the fixed target value. For $(*)$, we apply the IS technique, which states that $\mathbb{E}_{x\sim p}[f(x)] = \mathbb{E}_{x\sim q}\left[\frac{p(x)}{q(x)}f(x)\right]$ for any probability distributions $p$ and $q$, and arbitrary function $f$, and $w_{s,a}^Q = \frac{\hat{\beta}Q(a|s)}{\hat{\beta}(a|s)} = \frac{\exp(Q(s,a))}{\mathbb{E}_{a'\sim\hat{\beta}(\cdot|s)}[\exp(Q(s,a'))]}$ is the importance sampling (IS) ratio between $\hat{\beta}Q$ and $\hat{\beta}$.

## B.2 Practical Implementation for EPQ

Our implementation basically follows the setup of CQL [21]. We use the Gaussian policy $\pi$ with a Tanh$(\cdot)$ layer proposed by Haarnoja et al. [4], and parameterize the policy $\pi$ and $Q$-function using neural network parameters $\phi$ and $\theta$, respectively. Then, we update the policy to maximize $Q_\theta$ with its entropy $\mathcal{H}(\pi_\phi) = \mathbb{E}_{\pi_\phi}[-\log \pi_\phi]$, following the maximum entropy principle [4] as explained in Section 3.3, to account for stochastic policies. Then, we can redefine the policy loss function $L(\pi)$ defined in (5) as the policy loss function $L_\pi(\phi)$ for policy parameter $\phi$, given by

$$L_\pi(\phi) = \mathbb{E}_{s\sim D,\, a\sim\pi_\phi}[-Q_\theta(s, a) + \log \pi_\phi(a|s)]. \tag{B.1}$$

For the $Q$-loss function in (4), we use the IS ratio $w_{s,a}^Q$ in (4) to account for prioritized sampling based on $\hat{\beta}^Q$. However, $\hat{\beta}^Q$ discards samples with low IS weights, which can reduce sample efficiency. To address this, we utilize the clipped IS weight $\max(c_{\min}, w_{s,a}^Q)$, where $c_{\min} \in (0, 1]$ is the IS clipping constant. This clipped IS weight is multiplied only to the term $(Q(s, a) - \mathcal{B}^\pi Q(s, a))^2$ in (4) to ensure that we can exploit all data samples for $Q$-learning while preserving the proposed penalty. The detailed analysis for $c_{\min}$ is provided in Appendix D. In addition, the optimal policy that maximizes (B.1) follows the Boltzmann distribution, proportional to $\exp(Q_\theta(s, \cdot))$. It has been proven in Kumar et al. [21] that the optimal policy satisfies $\mathbb{E}_{a\sim\pi}[Q_\theta(s, a)] + H(\pi) = \log \sum_{a\sim\mathcal{A}} \exp Q_\theta(s, a)$, so we can replace the $\mathbb{E}_{a'\sim\pi}[Q_\theta(s, a')]$ term in (4) with $\log \sum_{a'\sim\mathcal{A}} \exp Q_\theta(s, a')$, given that $H(\pi)$ does not depend on the $Q$-function. The Bellman operator $\mathcal{B}^\pi$ can be estimated by samples in the dataset as $\mathcal{B}^\pi Q_\theta \approx r(s, a) + \mathbb{E}_{a'\sim\pi}\gamma Q_{\bar{\theta}}(s', a')$, where $\bar{\theta}$ is the parameter of the target $Q$-function. The target network is updated using exponential moving average (EMA) with temperature $\eta_{\bar{\theta}} = 0.005$, as proposed in the deep Q-network (DQN) [52]. Finally, by applying IS clipping and $\log \sum_a \exp Q$ to the $Q$-loss function (4) and redefining it as the value loss function for the value parameter $\theta$, we obtain the following refined value loss function $L_Q(\theta)$ as follows:

$$L_Q(\theta) = \frac{1}{2}\mathbb{E}_{s,a,s'\sim D}\big[\max(c_{\min}, w_{s,a}^Q) \cdot (r(s, a) + \mathbb{E}_{a'\sim\pi}\gamma Q_{\bar{\theta}}(s', a') - Q_\theta(s, a))^2\big] \tag{B.2}$$

$$+ \alpha\mathbb{E}_{s,a\sim D}\left[w_{s,a}^Q f_\tau^{\pi,\hat{\beta}}(s)\left(\log \sum_{a'\in\mathcal{A}} Q_\theta(s, a') - Q_\theta(s, a)\right)\right],$$

where $\hat{\beta}$ is pre-trained by behavior cloning (BC) [18, 53] to compute $f_\tau^{\pi,\hat{\beta}}$. The parameters $\phi$ and $\theta$ are updated to minimize their loss functions $L_\pi(\phi)$ and $L_Q(\theta)$ with learning rate $\eta_\phi$ and $\eta_\theta$, respectively. Detailed implementations for estimating the behavior policy $\hat{\beta}$, the IS weight $w_{s,a}^Q$, and $\log \sum_a \exp Q$ are provided in Appendices B.3, B.4, and B.5, respectively.

## B.3 Behavior Policy Estimation Based on Variational Auto-Encoder

In Section B.2, we estimate the behavior policy $\beta$ that generates the data samples in $D$ necessary for calculating the penalty adaptation factor $f_\tau^{\pi,\hat{\beta}}$ in equation (2). To estimate the behavior policy $\hat{\beta}$, we employ the variational auto-encoder (VAE), one of the most representative variational inference methods, to approximate the underlying distribution of a large dataset based on the variational lower bound [53]. In the context of VAE, we define an encoder model $p_\psi(z|s, a)$ and a decoder model $q_\psi(a|z, s)$ parameterized by $\psi$, where $z$ is the latent variable whose prior distribution $p(z)$ follows the multivariate normal distribution, i.e., $p(z) \sim N(0, I)$. Assuming independence among all data samples, we can derive the variational lower bound for the likelihood of $\hat{\beta}$ as proposed by Kingma and Welling [53]:

$$\log \beta(a|s) \geq \underbrace{\mathbb{E}_{z\sim p_\psi(\cdot|s,a)}[\log q_\psi(a|z, s)] - D_{KL}(p_\psi(z|s, a)||p(z))}_{\text{the variational lower bound}}, \quad \forall s, a \in D \tag{B.3}$$

where $D_{KL}(p||q) = \mathbb{E}_p[\log p - \log q]$ is the Kullback-Leibler (KL) divergence between two distributions $p$ and $q$. In this paper, since we consider the deterministic decoder $q_\psi(z, s)$, the formal term $\mathbb{E}_{z\sim p_\psi(\cdot|s,a)}[\log q_\psi(a|z, s)]$ can be replaced with the mean square error (MSE) as $\mathbb{E}_{z\sim p_\psi(\cdot|s,a)}[\log q_\psi(a|z, s)] \approx \mathbb{E}_{z\sim p_\psi(\cdot|s,a)}[(q_\psi(z, s) - a)^2]$. At each $k$-th iteration, we update the parameter $\psi$ of VAE to maximize the lower bound in equation (B.3). The $\log \beta$ can be estimated using the variational lower-bound in (B.3) to obtain $f_\tau^{\pi,\hat{\beta}}$. The hyperparameter setup for the VAE is provided in Table 2.

Table 2: Hyperparameter setup for VAE

| **VAE Hyperparameters** | |
| --- | --- |
| $z$ dimension | 2· state dimension |
| Hidden activation function | ReLU Layer |
| Encoder network $p_\psi$ | (512, $2 \cdot z$ dim.) |
| | (512,512) |
| | (state dim. + action dim., 512) |
| Decoder network $q_\psi$ | (512, action dim.) |
| | (512,512) |
| | ($z$ dim. + state dim., 512) |

## B.4   Implementation of IS Weight $w_{s,a}^Q$

In order to consider the prioritized data distribution $\hat{\beta}^Q$ proposed in Section 3.3, we use the importance sampling (IS) weight defined by

$$w_{s,a}^Q = \frac{\hat{\beta}^Q(a|s)}{\hat{\beta}(a|s)} = \frac{\exp(Q(s,a))}{\mathbb{E}_{a' \sim \hat{\beta}(\cdot|s)}[\exp(Q(s,a'))]}, \ \forall s, a \in D. \tag{B.4}$$

Since the computation of $\mathbb{E}_{a' \sim \hat{\beta}(\cdot|s)}$ makes it difficult to know the exact possible action set for state $s$, we approximately estimate the IS weight based on clustering as follows:

$$w_{s,a}^Q = \frac{\exp(Q(s,a))}{\mathbb{E}_{a' \sim \hat{\beta}(\cdot|s)}[\exp(Q(s,a'))]} \approx \frac{\exp(Q(s,a))}{\frac{1}{|\mathcal{C}_{s,a}|} \sum_{(s',a') \in \mathcal{C}_{s,a}} \exp(Q(s',a'))}, \ \forall s, a \in D. \tag{B.5}$$

Here, $\mathcal{C}_{s,a}$ is the cluster that contains data samples adjacent to $(s,a)$, defined by

$$\mathcal{C}_{s,a} = \{(s',a') \in D \mid ||s - s'||_2 \le \epsilon \cdot \bar{d}_{\text{closest}}\}, \tag{B.6}$$

where the cluster $\mathcal{C}_{s,a}$ can be directly obtained using the nearest neighbor (NN) algorithm [54] provided in the Python library. $\epsilon \cdot \bar{d}_{\text{closest}}$ is the radius of the cluster, and $\bar{d}_{\text{closest}}$ is the average distance between the closest states from each task. In our implementation, we control the radius parameter $\epsilon > 0$ to adjust the number of adjacent samples for the estimation of IS Weight $w_{s,a}^Q$. In addition, using the $Q$-function in the IS weight term makes the learning unstable since the $Q$-function continuously changes as the learning progresses. Thus, instead of the $Q$-function, we use the regularized empirical return $G_t/\zeta$ for each state-action pair obtained by the trajectories stored in $D$, where $\zeta > 0$ is the regularizing temperature. Upon the increase of $\zeta$, the returned difference between adjacent samples in the cluster decreases, so the effect of prioritization can be reduced. The detailed analysis for $\epsilon$ and $\zeta$ is provided in Appendix D.

## B.5 Implementation of $Q$-loss Function

In equation (B.2), the final $Q$-loss function of proposed EPQ is given by

$$L_Q(\theta) = \frac{1}{2}\mathbb{E}_{s,a,s'\sim D}\Big[\max(c_{\min}, w_{s,a}^Q)\left(r(s,a) + \mathbb{E}_{a'\sim\pi}\gamma Q_{\bar{\theta}}(s',a') - Q_\theta(s,a)\right)^2\Big]$$

$$+ \alpha\mathbb{E}_{s,a\sim D}\left[w_{s,a}^Q f_\tau^{\pi,\hat{\beta}}(s)\left(\log\sum_{a'\in\mathcal{A}}\exp Q_\theta(s,a') - Q_\theta(s,a)\right)\right].$$

Here, we can estimate $\log\sum_a \exp Q(s,a)$ based on the method proposed in CQL [21] as follows:

$$\log\sum_a \exp Q(s,a) = \log\left(\frac{1}{2}\sum_a \pi(a|s)\{\exp(Q(s,a) - \log\pi(a|s))\} + \frac{1}{2}\sum_a \rho_d\{\exp(Q(s,a) - \log\rho_d)\}\right)$$

$$\approx \log\left(\frac{1}{2N_a}\sum_{a_n\sim\pi}^{N_a}(\exp(Q(s,a_n) - \log\pi(a_n|s))) + \frac{1}{2N_a}\sum_{a_n\sim\text{Unif}(\mathcal{A})}^{N_a}(\exp(Q(s,a_n) - \log\rho_d)))\right),$$

$$\tag{B.7}$$

where $N_a$ is the number of action sampling, $\text{Unif}(\mathcal{A})$ is a Uniform distribution on $\mathcal{A}$, and $\rho_d$ is the density of uniform distribution.

## B.6 Time comparison with other offline RL methods

In this sectrion, we compare the runtime of EPQ with other baseline algorithms: CQL, Onestep, IQL, MCQ, and MISA in Table 3 below. For a fair comparison across all algorithms, we conducted experiments on the Hopper-medium task, which is a popular dataset for comparing computational costs [48, 55], on a single server equipped with an Intel Xeon Gold 6336Y CPU and one NVIDIA RTX A5000 GPU. We measured both epoch runtime during 1,000 gradient steps and score runtime that each algorithm takes to achieve certain normalized scores.

From the epoch runtime results in Table 3, we can observe that EPQ takes approximately 2-30% more runtime per gradient step compared to the CQL baseline. Note that Onestep RL may seem to have very short execution time compared to other algorithms, but one must consider the significantly longer pretraining time required to learn the $Q$-function of behavior policy accurately. Additionally, compared to faster offline RL algorithms such as IQL and MISA, EPQ requires more runtime per step and exhibits a similar runtime to MCQ, another conservative Q-learning algorithm. However, according to the score runtime results in Table 3, we can observe that only proposed EPQ achieves a score of 100 points, while all other algorithms fail to reach this score. Particularly, compared to MCQ, which also considers CQL as a baseline, EPQ achieves the same score with significantly less runtime. Therefore, while EPQ may consume slightly more runtime per gradient step compared to other algorithms, we can conclude that proposed EPQ offers substantial advantages in terms of convergence performance over other algorithms.

Table 3: Runtime comparison: Epoch runtime and Score runtime

| epoch runtime(s) | CQL | Onestep | IQL | MCQ | MISA | EPQ |
|---|---|---|---|---|---|---|
| 1,000 gradient steps | 43.1 | 12.6 | 13.8 | 58.1 | 23.5 | 54.8 |

| score runtime(s) Normalized average return | CQL | Onestep | IQL | MCQ | MISA | EPQ |
|---|---|---|---|---|---|---|
| 60 | 3540.0 | 252.5 | 1600.2 | 31,143.4 | 4,632.7 | 3,232.2 |
| 80 | - | 568.4 | - | 49,359.7 | - | 21,920.0 |
| 100 | - | - | - | - | - | 30,633.2 |

# C Hyperparameter Setup

The implementation of proposed EPQ basically follows the implementation of the CQL algorithm [21]. First, we provide the details of the shared algorithm hyperparameters in Table 4. In Table 4, we compare the shared algorithm hyperparameters of CQL, the revised version of CQL (revised), and proposed EPQ. CQL (revised) considers the same hyperparameter setup with our algorithm for Adroit tasks since the reproduced performance of CQL (reprod.) using the author-provided hyperparameter setup significantly underperforms compared to the result of CQL (paper) in Table 1.

For the coefficient of entropy term in the policy update (B.1), CQL automatically controls the entropy coefficient so that the entropy of $\pi$ goes to the target entropy, as proposed in Haarnoja et al. [56]. We observe that while the automatic control of policy entropy proves effective for Mujoco tasks, it adversely affects the performance in Adroit tasks since a policy with low entropy can lead to significant overestimation errors in Adroit tasks. Thus, we considered fixed entropy coefficient for Adroit tasks as in Table 4. In addition, CQL controls the penalizing constant $\alpha$ based on Lagrangian method [21] for Adroit tasks, but we also observe that the automatic control of $\alpha$ destabilizes training, leading to poor performance. Therefore, we considered fixed penalizing constant for Adroit tasks in Table 4 for stable learning.

In addition, in Table 5, we provide the details of the task hyperparameters regarding our contributions in the proposed EPQ: the penalty control threshold $\tau$ and the IS clipping factor $c_{\min}$ in the $Q$-loss implementation in (B.2), and the cluster radius $\epsilon$ and regularizing temperature $\zeta$ for the practical implementation of IS clipping factor $w_{s,a}^Q$ in Section B.4. Note that $\rho$ in Table 5 represents the log-density of uniform distribution. For the task hyperparameters, we consider various hyperparameter setups and provide the best hyperparameter setup for all considered tasks in Table 5. The results are based on the ablations studies provided in Section 4.3 and Appendix D.

Table 4: Algorithm hyperparameter setup of CQL, CQL (revised), and EPQ (ours) algorithms

| Hyperparameters | CQL | CQL (revised) (for Adroit) | EPQ |
|---|---|---|---|
| Policy learning rate $\eta_\phi$ | 1e-4 | 1e-4 | 1e-4 |
| Value function learning rate $\eta_\theta$ | 3e-4 | 3e-4 | 3e-4 |
| Soft target update coefficient $\eta_{\bar\theta}$ | 0.005 | 0.005 | 0.005 |
| Batch size | 256 | 256 | 256 |
| The number of sampling $N_a$ | 10 | 10 | 10 |
| Initial behavior cloning steps | 10000 | 10000 | 10000 |
| Gradient steps for training | 3m (0.3m for Adroit) | 0.3m | 3m (0.3m for Adroit) |
| Entropy coefficient $\eta_\theta$ | Auto | 0.5 | Auto (0.5 for Adroit) |
| Penalizing constant $\alpha$ | Auto (10 for MuJoCo) | 5 or 20 | 20 for MuJoCo
5 or 20 for Adroit
5 or Auto for AntMaze |
| Discount factor $\gamma$ | 0.99 | 0.9 or 0.95 | 0.99 (0.9 or 0.95 for Adroit) |

Table 5: Task hyperparameter setup for Mujoco tasks and Adroit tasks

| **Mujoco Tasks** | $\tau/\rho$ | $c_{\min}$ | $\epsilon$ | $\zeta$ |
|---|---|---|---|---|
| halfcheetah-random | 10 | 0.2 | 2 | 2 |
| hopper-random | 2 | 0.1 | 0.5 | 2 |
| walker2d-random | 1 | 0.2 | 2 | 0.5 |
| halfcheetah-medium | 10 | 0.2 | 0.5 | 2 |
| hopper-medium | 0.2 | 0.5 | 2 | 5 |
| walker2d-medium | 1 | 0.5 | 2 | 2 |
| halfcheetah-medium-expert | 1.0 | 0.2 | 0.5 | 2 |
| hopper-medium-expert | 1 | 0.2 | 0.5 | 2 |
| walker2d-medium-expert | 1.0 | 0.2 | 0.5 | 2 |
| halfcheetah-expert | 1 | 0.2 | 0.5 | 2 |
| hopper-expert | 1 | 0.2 | 0.5 | 2 |
| walker2d-expert | 0.5 | 0.2 | 2.0 | 2 |
| halfcheetah-medium-replay | 2 | 0.2 | 0.5 | 2 |
| hopper-medium-replay | 2 | 0.2 | 0.5 | 2 |
| walker2d-medium-replay | 0.2 | 0.5 | 1.0 | 2 |
| halfcheetah-full-replay | 1.5 | 0.2 | 0.5 | 2 |
| hopper-full-replay | 2.0 | 0.2 | 1.0 | 2 |
| walker2d-full-replay | 1.0 | 0.2 | 0.5 | 2 |
| **Adroit Tasks** | $\tau/\rho$ | $c_{\min}$ | $\epsilon$ | $\zeta$ |
| pen-human | 0.05 | 0.5 | 1.0 | 200 |
| door-human | 0.05 | 0.5 | 0.5 | 200 |
| hammer-human | 0.1 | 0.2 | 5 | 100 |
| relocate-human | 0.2 | 0.2 | 2 | 10 |
| pen-cloned | 0.2 | 0.2 | 5 | 50 |
| door-cloned | 0.2 | 0.5 | 1 | 10 |
| hammer-cloned | 0.2 | 0.2 | 5 | 100 |
| relocate-cloned | 0.2 | 0.2 | 5 | 10 |
| **AntMaze Tasks** | $\tau/\rho$ | $c_{\min}$ | $\epsilon$ | $\zeta$ |
| umaze | 10 | 0.2 | 2 | 2 |
| umaze-diverse | 10 | 0.2 | 2 | 2 |
| medium-play | 0.1 | 0.2 | 1 | 2 |
| medium-diverse | 0.1 | 0.2 | 1 | 2 |
| large-play | 0.1 | 0.2 | 1 | 2 |
| large-diverse | 0.1 | 0.2 | 1 | 2 |

# D   Additional Ablation Studies Related to $w_{s,a}^Q$ Estimation

In this section, we provide additional ablation studies related to IS weight $w_{s,a}^Q$ estimation in Appendix B. For analysis, Fig. 8 shows the performance plot when the IS clipping factor $c_{\min}$, the cluster radius $\epsilon$, and the temperature $\zeta$ change.

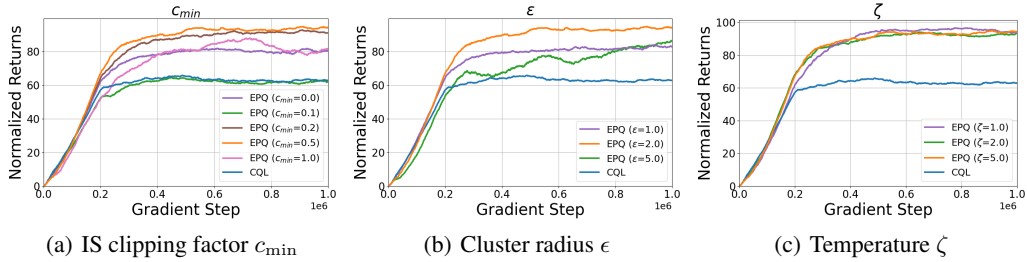

|  (a) IS clipping factor $c_{\min}$  |  (b) Cluster radius $\epsilon$  |  (c) Temperature $\zeta$  |

Figure 8: Additional ablation studies on Hopper-medium task

**IS Clipping Factor** $c_{\min}$: In the EPQ implementation, the IS clipping factor $c_{\min}$ is employed to clip the IS weight $w_{s,a}^Q$ to prevent the exclusion of data samples with relatively low $w_{s,a}^Q$. When $c_{\min} = 0$, low-quality samples with low $w_{s,a}^Q$ are not utilized at all based on the prioritization in Section 3.3. However, as $c_{\min}$ increases, these low-quality samples are increasingly exploited. Fig. 7(c) illustrates the performance of EPQ with varying $c_{\min}$, and EPQ achieves the best performance when $c_{\min} = 0.5$. This result suggests that it is more beneficial to use low-quality samples with proper priority rather than discarding them entirely.

**Cluster Radius** $\epsilon$: As explained in Appendix B.4, we can control the number of adjacent samples in the cluster based on the radius $\epsilon$. From the results illustrated in Fig. 8(a), we can observe that EPQ with $d = 2.0$ performs best, and a decrease or an increase in $\epsilon$ can significantly affect the performance indicating that $\epsilon$ must be chosen properly for each task to find the cluster that contains adjacent samples appropriately. If $\epsilon$ is too small, the cluster will hardly contain adjacent samples, and if $\epsilon$ is too large, samples that should be distinguished will aggregate in the same cluster, adversely affecting the performance.

**Temperature** $\zeta$: As proposed in Section 3.3, samples in the dataset are prioritized according to the definition of $w_{s,a}^Q$. Since the samples with higher $Q$ values are more likely to be selected for the update of the $Q$-function, temperature $\zeta$ controls the amount of prioritization, as explained in Appendix B.4. Increasing $\zeta$ reduces the difference in the $Q$-function between the samples, putting less emphasis on prioritization. Fig. 8(b) shows the performance change according to the change in $\zeta$, where the results state that the performance does not heavily depend on $\zeta$. From the ablation study, we can conclude that the radius $\epsilon$ has a greater influence on the performance of Hopper-medium task compared to the temperature $\zeta$.

# E    Additional Performance Comparison on Adroit Tasks

For adroit tasks, the performance of CQL (reprod.) is too low compared to CQL (paper) in Table 1, so we additionally provide the performance result of the revised version of CQL provided in Section C. We also compare the performance of EPQ with the performance of CQL (revised) on various adroit tasks, and Table 6 shows the corresponding comparison results. From the result, we can see that CQL (revised) greatly enhances the performance of CQL on adroit tasks, but EPQ still outperforms CQL (revised), which demonstrates the intact advantage of the proposed exclusive penalty and prioritized dataset well on the adroit tasks.

Table 6: Performance comparison of CQL (paper), CQL (revised), and EPQ (ours) on Adroit tasks.

| Task | CQL (paper) | CQL (revised) | EPQ |
|------|-------------|---------------|-----|
| pen-human | 55.8 | 82.0±6.2 | **83.9±6.8** |
| door-human | 9.1 | 7.8±0.5 | **13.2 ± 2.4** |
| hammer-human | 2.1 | **6.4±5.4** | 3.9±5.0 |
| relocate-human | **0.4** | 0.1±0.2 | **0.3±0.2** |
| pen-cloned | 40.3 | **90.7±4.8** | **91.8±4.7** |
| door-cloned | 3.5 | 1.3±2.2 | **5.8±2.8** |
| hammer-cloned | 5.7 | 2.0±1.3 | **22.8±15.3** |
| relocate-cloned | -0.1 | 0.0±0.0 | **0.1±0.1** |
| **Adroit Tasks Total** | 116.8 | 190.3 | **221.8** |

# F    Limitations

The proposed EPQ significantly improves performance over the existing CQL baseline on various D4RL tasks, but there are many hyperparamaters that need to be optimized. We newly consider the penalty control threshold $\tau$, IS clipping factor $c_{\min}$, the cluster radius $\epsilon$, and the regularizing temperature $\zeta$. Therefore, in order for the proposed EPQ to perform well, it is necessary to find the optimal performance by considering various hyperparameter setup, which may require some interaction with the environment.

# G    Broader Impact

Nevertheless, in real-world situations, engaging with the environment can be costly. Particularly in high-risk contexts such as disaster scenarios, acquiring adequate data for learning can be quite challenging. Our research is primarily focused on offline settings and we present a novel method, EPQ, holds the potential for practical applications in real-life situations where the interaction is not available, and exhibits promise in addressing the challenges posed by offline RL algorithms. Consequently, our work carries several potential societal implications, although we believe that none require specific emphasis in this context.

