# OpenReview forum: "Exclusively Penalized Q-learning for Offline Reinforcement Learning"
_NeurIPS.cc/2024/Conference — NeurIPS 2024 spotlight_

### Official Review · Reviewer_Hh4A · 2024-06-21

**Soundness:** 2
**Presentation:** 3
**Contribution:** 2
**Rating:** 5
**Confidence:** 4

**Summary:**

This paper investigates an important problem in offline reinforcement learning (RL), say mitigating unnecessary conservatism in value function. The authors achieve this by selectively penalizing states that are prone to inducing estimation errors, i.e., $f$. The core idea is to train an exclusive penalty $P_\tau=f_\tau^{\pi,\hat{\beta}}(s) \left(\dfrac{\pi}{\hat{\beta}}-1\right)$, where $f_\tau^{\pi,\hat{\beta}}(s)$ is the penalty adaptation factor. $f_\tau^{\pi,\hat{\beta}}(s)$ assigns smaller weights on in-distribution transitions and a larger weight on out-of-distribution (OOD) transitions. Furthermore, the authors propose the *Prioritized Dataset* (PD) trick to reduce unnecessary bias. Based on PD, the authors derive the final optimization objective of their EPQ approach. The authors conduct experiments on numerous D4RL datasets and show that EPQ can outperform previous methods on many datasets.

**Strengths:**

I appreciate that the authors set their focus on value-based offline RL methods and propose an interesting weighting method for mitigating the over-conservatism phenomenon in CQL, akin to mildly conservative Q-learning (MCQ) algorithm. Many recent offline RL algorithms study policy regularization approaches and somewhat neglect the advances of learning offline policies purely from a value function optimization perspective. The overall method is, as far as the reviewer can tell, novel. Though the proposed EPQ method can be seen as one variant of CQL, the proposed method is interesting and can address the unnecessary conservatism on in-distribution data.

The presentation of this paper is very nice and I personally quite like it. The authors include many toy examples (e.g., Figure 3, Figure 4) and illustrations of their method (Figure 2). This is of great help in aiding the readers to quickly capture the key points that the authors would like to convey. The authors also compare their method against numerous strong offline RL value-based and policy regularization algorithms and demonstrate that EPQ exhibits quite strong performance on many datasets.

**Weaknesses:**

This paper has the following potential drawbacks

- EPQ introduces many hyperparameters, and one needs to manually find the optimal ones on a new dataset. This can impede the practical application of EPQ in real-world problems

- The ablation study part is insufficient. The authors only conduct experiments on one single environment in the main text and the appendix (see Figure 6 and Figure 7). This ought to be evaluated on wider datasets to thoroughly examine the hyperparameter sensitivity and how different components/hyperparameters affect the performance of EPQ. The influence of the PD trick also should be investigated on wider datasets. Based on Figure 6, it turns out that EPQ with PD and EPQ without PD exhibit similar performance.

- Equation 3 depicts that the target value is *corrected* by the introduced exclusive penalty term $P_\tau$. The final objective (Equation 4) is also derived based on this. Equation 3 reminds me of the RND-series in offline RL, e.g., SAC-RND [1], and SAC-DRND [2]. Typically, they also subtract a penalty term from the target value to pursue conservatism and mitigate overestimation. I think $P_\tau$ can also be viewed as a similar role. Any comments here? Are there any advantages of the introduced penalty term $P_\tau$ over the penalty term given by RND? How these methods can be connected? What if we directly optimize Equation 3 and tune $\alpha$?

[1] Anti-Exploration by Random Network Distillation. ICML 2023

[2] Exploration and Anti-Exploration with Distributional Random Network Distillation. ICML 2024.

**Questions:**

I have the following questions

- Is there a reason that you do not report the standard deviation of EPQ on halfcheetah-expert dataset in Table 1?

- it seems CQL($\alpha=0.0$) in Figure 5(a) middle early stops, it there a reason for this?

- how do you expect EPQ to be applied in real-world tasks? Any suggestions or instructions on how to tune the introduced hyperparameters?

Some minor points:

- I noticed that one cannot be directed to the corresponding images/tables by clicking, e.g. Fig 2. This problem hinders the smooth reading and understanding of the content and should be fixed in the revision

- The referenced figures/tables should be distinguished, e.g., Line 90 *Fig. 1(a) and (b)* should be *Fig. 1(a) and 1(b)*. Please check the manuscript and fix all such types of issues.

- Equation 5, $Q\_( s,a)$ ==> $Q(s,a)$

- sometimes the authors just write *adroit* instead of *Adroit* (e.g., Line 232)

- the captions in Figure 5 and Figure 6 are too simple

**Limitations:**

The authors include an honest discussion part on the limitations of their work in the appendix. I personally agree with that.

---

> ### Author Rebuttal · Authors · 2024-08-06
>
> Thank you for your valuable feedback. We have addressed the feedback regarding Fig. 2 and conducted additional ablation studies to answer the reviewer's concerns related to the proposed prioritized dataset (PD) and the penalty control threshold $\tau$, as detailed in our global response. In response to other concerns raised by the reviewers, we provide the following author response.
>
> **Practical application of EPQ**
>
> We understand the reviewer's concern about the practical application of EPQ, given that tuning many hyperparameters for each environment can be costly. However, as the experimental results in Appendix D suggest, EPQ demonstrates plausible performance even with non-optimal parameters. Additionally, as the reviewer suggested, we conducted additional experiments to test the sensitivity EPQ on our main parameter $\tau$ in Halfcheetah-medium and Hopper-random environments, and reported these findings in Fig. R.3. As explained in the global response, while the optimal $\tau$ varies depending on the variety of possible actions in each task, EPQ showed consistent performance within a certain range of $\tau$ values. Notably, in the Hopper-random environment, EPQ achieved superior performance across a range of $\tau/\rho \in [0.2, 0.5, 1.0, 2.0]$. This suggests that precise tuning of all parameters may not be necessary, as EPQ performs well with a reasonable range of parameter settings.
>
>
> **Distinction to intrinsic reward based offline RL**
>
> As the reviewer pointed out, many penalty-based offline RL methods apply penalties to the target values to reduce the estimation error of policy actions not present in the dataset [1, 2, 3]. Unlike SAC-RND [1], which uses random network distillation as an uncertainty estimator for penalizing uncertain actions, our approach adjusts the penalty based on whether the actions being evaluated are sufficiently represented in the dataset. This allows for a theoretical analysis of the causes of estimation bias and the appropriate degree of penalty required to mitigate it. Additionally, we observed that directly optimizing Eq. (3) in the paper, which imposes the penalty as an intrinsic reward, can lead to unstable learning due to the large penalty continuously fluctuating the target values. In contrast, our approach ensures stable learning by directly applying the penalties to the $Q$-values, thereby enabling the superior performance of EPQ.
>
> **Clarity issues for figures in the paper**
>
> **Fig. 2:** We have regarded the reviewer's concerns on the clarity of Fig. 2 in our global response. Please refer to it for our detailed response.
>
> **Table 1:** The exclusion of the standard deviation of EPQ on Halfcheetah-expert dataset is a typo. The experimental results showed  $107.2 \pm 0.2$. Thank you for pointing it out.
>
> **Early stopped result in Fig. 5:** In the experiment shown in Fig. 5(a), the estimation bias in CQL with $\alpha=0$ became excessively large, causing the gradients to explode and resulting in forced termination of the training. In fact, Fig. 5(a) shows that the estimation bias reached up to $1e^{13}$. Therefore, we reported the results in Fig. 5 up to the point where the experiment was terminated.
>
> **Minor issues**
>
> We will address the reviewer's feedback by correcting typos, distinguishing figure references, and enhancing captions to improve the overall presentation of the paper.
>
>
> [1] Nikulin, Alexander, et al. "Anti-exploration by random network distillation." International Conference on Machine Learning. PMLR, 2023.
>
> [2] Kumar, Aviral, et al. "Conservative q-learning for offline reinforcement learning." Advances in Neural Information Processing Systems 33, 2020.
>
> [3] Kostrikov, Ilya, Ashvin Nair, and Sergey Levine. "Offline reinforcement learning with implicit q-learning." arXiv:2110.06169, 2021.

---

> > ### Comment · Reviewer_Hh4A · 2024-08-11
> >
> > Thanks for the clarifications. It would be good if the authors could incorporate my suggestions (e.g., ablation study, early stop issue) when preparing the camera-ready manuscript. The main drawback of this paper is that EPQ introduces many hyperparameters, and one needs to manually find the optimal ones on a new dataset, as commented. However, since existing methods like ReBARC also require significant hyperparameter search to achieve good performance, I think it is okay and believe this paper can be accepted. I would vote for acceptance.

---

> > > ### Author Response · Authors · 2024-08-12
> > >
> > > Thank you for the prompt response. We are glad to hear that our answers have helped in understanding the paper. If you have any additional questions or need further information, please let us know.

---

### Official Review · Reviewer_eWrQ · 2024-06-28

**Soundness:** 3
**Presentation:** 2
**Contribution:** 3
**Rating:** 7
**Confidence:** 4

**Summary:**

This paper showed that the popular conservative Q-learning introduces bias into the Q function by its restrictive penalty. It proposed to enforce an adaptive penalty term based on the dataset to avoid bias when the dataset support disagrees with the learned policy. Experimental results supported the authors' claims that the proposed method can effectively reduce estimation bias and outperform many baselines.

**Strengths:**

Conservatism is common in offline RL methods. CQL is one of the most popular algorithms that enforces the conservatism by penalizing its Q value, and hence it runs the risk of introducing bias. This paper nicely illustrated a possible source of bias originating from the difference between dataset actions and policy actions, and proposed to correct the bias by the penalty adaptation factor. The authors justified the proposed penalty with both pedagogical examples and more challenging problems.

**Weaknesses:**

While the paper is well-written in general, I believe the presentation can be further improved by adding making the exlanation more accesible, especially Figure 1 and 2. The two figures served a pivotal role in illustrating the downside of CQL and the core idea of the proposed EPQ. Figure 1 would benefit from separating the information into two parts: (1) overlaps of the dataset/policy actions and (2) estimation bias of CQL under different $\alpha$.

**Questions:**

1. In Figure 2 the authors suggested the relationship
> let $\tau_2 < \tau_1$ when $N_1 < N_2$\

However, it seems there is chance that if $\tau_2$ is only slightly smaller than $\tau_1$, then $\tau_2$ would only be penalizing $\pi_3$ and not covering $\pi_1, \pi_2$. More generally, this relates to choosing an appropriate threshold. In the experiments the authors swept $(c\cdot \rho, c\in[0, 10])$, but what was the observation or rule of thumb here to make sure the threshold is meaningful?

2. Another question concerns the implementation. It seems the empirical behavior policy $\hat{\beta}(a|s)$ plays a crucial role in the adaptive penalty. In continuous domains the offline datasets often comprise only a single copy for any action, making the estimation of $\hat{\beta}(a|s) = \frac{N(s,a)}{N(s)}$ rather hard and inaccurate. Therefore, existing work like [1, 2] often use sophisticated models to estimate $\hat{\beta}(a|s)$ with the hope that it would generalize better. By contrast, it is surprising to see that Algorithm 1 used only simple BC and that was sufficient to fuel the superior performance of EPQ. What could be the reason?

References:\
[1] Mildly Conservative Q-Learning for Offline Reinforcement Learning
[2] Supported Policy Optimization for Offline Reinforcement Learning

**Limitations:**

Potential negative societal impact does not apply and the authors have discussed some of the limitations in the appendix.

---

> ### Author Rebuttal · Authors · 2024-08-06
>
> Thank you for your valuable feedback. We have addressed the feedback regarding Fig. 2 and conducted additional ablation studies to answer the reviewer's concerns related to the proposed prioritized dataset (PD) and the penalty control threshold $\tau$, as detailed in our global response. In response to other concerns raised by the reviewers, we provide the following author response.
>
> **Clarity issues for figures in the paper**
>
> **Fig. 1:** Although we could not include the revision of Fig. 1 due to limited space, we will improve the clarity of the paper by separating the illustration of CQL's estimation bias according to various penalizing constant $\alpha$ and the distributions of $\pi$ and $\hat{\beta}$, as suggested by the reviewer.
>
> **Fig. 2:** We have regarded the reviewer's concerns on the clarity of Fig. 2 in our global response. Please refer to it for our detailed response.
>
>
> **Choice of the penalty control threshold $\tau$**
>
> As noted by the reviewer, it is crucial to select an appropriate $\tau$ to avoid the situation described by the reviewer. Fig. 2 and R.1 suggest that, "with a fixed data distribution", as the number of datasets $N$ increases, actions are sampled more diversely, which implies that $\tau$ should be lower. However, in actual experimental environments, since the data distribution varies across environments and states, these factors need to be additionally considered for choosing the optimal $\tau$.
>
> To address this issue, we provide additional ablation studies regarding $\tau$ and the rule for selecting the threshold $\tau$ are detailed in the global response. Based on the experimental results provided for the global response, the overestimation error arises from samples that have not been visited, so the optimal $\tau$ tends to be determined by how diversely actions are sampled in the dataset for each environment. According to our analysis related to $\tau$, choosing a low threshold $\tau$ is effective when the dataset contains sufficiently diverse state-action pairs, whereas a higher threshold is preferable when the dataset is less comprehensive.
>
> **Behavior cloning**
>
> As the reviewer pointed out, predicting the empirical behavior policy $\hat{\beta}$ significantly impacts the performance of EPQ, especially since our penalty adaptation factor $f^{\pi,\hat{\beta}}_\tau$ is calculated using samples $D$ generated from the predicted behavior policy. Predicting the behavior policy in continuous space is challenging, and various models have been employed for this task, as the reviewer mentions. Instead of using simple behavior cloning (BC), we utilize a prediction model based on the variational lower bound [1], one of the most representative variational inference methods. As described in Appendix B.3, assuming independence among all data samples, the variational lower bound for the likelihood of $\beta$ can be derived as $\log\beta(a|s) \geq \mathbb{E}\_{z\sim p\_\psi(\cdot|s,a)}[\log q\_\psi(a|z,s)]- D\_{KL}(p\_\psi(z|s,a)||p(z)),~\forall s,a \in D$, where $p\_\psi (z|s,a)$ is an encoder model and $q\_\psi(a|z,s)$ a decoder model parameterized by $\psi$, and $z$ is the latent variable whose prior distribution $p(z)$ follows the multivariate normal distribution, i.e., $p(z)\sim N(0,I)$. Please refer to Appendix B.3 for a detailed explanation. By employing this variational approach, we obtained a more precise estimate of the behavior policy $\hat{\beta}$ compared to using simple BC.
>
> [1] Kingma, Diederik P., and Max Welling. "Auto-encoding variational bayes." arXiv:1312.6114, 2013.

---

> > ### Comment · Reviewer_eWrQ · 2024-08-09
> > **thanks for the response**
> >
> > Thanks for the detailed response. I think it makes sense for the new version to include the description posted here for choosing a suitable $\tau$. The authors have addressed my questions satisfactorily. I have raised my score to 7.

---

> > > ### Author Response · Authors · 2024-08-09
> > >
> > > Thank you for the prompt response and valuable feedback. We're glad that our answers have been helpful for clarification. Please let us know if you have any additional questions or need further information.

---

### Official Review · Reviewer_RxmU · 2024-07-03

**Soundness:** 4
**Presentation:** 4
**Contribution:** 4
**Rating:** 8
**Confidence:** 4

**Summary:**

The paper studies the problem of value estimation bias mitigation in offline reinforcement learning. Specifically, the paper takes the well-known Conservative Q-Learning algorithm as a starting point and improves its penalization scheme with an alternative one that applies provably less value underestimation bias. The paper reports results on the well-known D4RL benchmark and compares against alternative approaches.

**Strengths:**

* The paper points to a key problem in model-free offline reinforcement learning, i.e. model underfit caused by overconservatism due to excessive penalty application.

 * The paper follows a solid methodology to the problem that starts from illustrative data-driven toy results, continues with theoretical analysis, and ends up with a well-understood and well-justified improvement to an established algorithm

* The shown results are particularly comprehensive and strong.

* The presentation of the paper at stellar level. It is extremely easy to follow the story line, even though it is technically quite dense.

* The paper addresses the related literature well.

**Weaknesses:**

The paper builds its whole problem statement and solution on a big assumption: The environment dynamics will not be learned. In other terms, the problems it highlights and addresses are specific to the model-free offline RL approaches, although learning an environment model is not such a major limitation in the offline setting, where the assumption is that data and compute sources are generous, training time is also not an issue, unlike the online MBRL setup. The paper seems to miss this key positioning element in problem formulation. Its presentation will improve if this choice is made more explicit and the impact of the proposed solution is presented accordingly.

Minor: I found Figure 2 extremely difficult to grasp. The authors may consider simplifying it a little and extending the caption with an explanation of it.

**Questions:**

* Is there a particular reason why the EDAC algorithm of An et al. [40] is not in the comparison list? It is also model free, reports results on the same benchmarks and it performs better than MISA on Mujoco Tasks, i.e. 85.2*18 = 1533.6.

 * Section 4.2 does not specify whether the estimation bias is compared between the predicted Q-value and the "discounted" observed return. Can the authors confirm that discounting has been applied in the results shown in Fig 6? By bare eye the bias looked to me too high.

**Limitations:**

The paper does not address its own limitations and does not specify the potential negative societal impact of the presented work.

---

> ### Author Rebuttal · Authors · 2024-08-06
>
> Thank you for your valuable feedback. We have addressed the feedback regarding Fig. 2 and conducted additional ablation studies related to the proposed prioritized dataset (PD) and the penalty control threshold $\tau$, as detailed in our global response. In response to other concerns raised by the reviewers, we provide the following author response.
>
> **Problem formulation in model-free offline RL**
>
> Model-free offline RL focuses on measuring and addressing overestimation bias in policies $\pi$ and solving distributional shift issues based on this measure, whereas model-based offline RL concentrates on how to mitigate distributional shift when learning from samples generated using a dynamics model. As a result, these two domains have different focuses and are both areas of active research. In our work, we considered the model-free setting to focus on how CQL, a model-free offline RL method, induces underestimation bias and how to address this issue. However, as noted by the reviewer, since model-based RL can lead to more efficient learning, extending our method to the model-based setting could be a valuable area for future work. We appreciate the reviewer’s suggestion and will actively consider it in our future research.
>
> **Comparison with EDAC**
>
> There are various approaches addressing the estimation bias of policy actions that may not be present in the dataset in offline RL setups. Our work specifically focuses on directly penalizing $Q$-functions to reduce overestimation, based on the analysis of estimation bias in $Q$-functions. Consequently, we prioritized comparisons with methods like CQL [1] and IQL [2], which share a similar focus on mitigating overestimation. On the other hand, EDAC [3] employs a clipped $Q$-learning method based on the confidence of $Q$-value predictions, presenting a different approach to addressing overestimation. Given that EDAC's methodology differs from ours, we opted to compare offline RL methods that reduce overestimation through penalty or constraint mechanisms, by examining the distributions of $\pi$ and $\beta$ from a methodological perspective rather than solely focusing on performance.
>
> However, we acknowledge the significance of EDAC in the offline RL domain, as highlighted in our Related Works section. In response to the reviewer’s comments, we will directly compare the performance of our proposed EPQ method with EDAC based on reported results for commonly considered environments in both EDAC and our study. For the Mujoco tasks, the average scores across all tasks were similar, with EPQ scoring 85.4 and EDAC scoring 85.2. In the Adroit tasks, however, EPQ demonstrated superior performance with an average score of 27.7 compared to EDAC’s average score of 17.7, showing nearly a twofold improvement. The Adroit task is particularly challenging due to its sparse rewards and limited variety of state-action pairs in the dataset, which makes learning difficult. Thus, EPQ effectively demonstrates its superiority over EDAC in such challenging scenarios.
>
> **Clarity issues for figures in the paper**
>
> **Fig. 2:** We have regarded the reviewer's concerns on the clarity of Fig. 2 in our global response. Please refer to it for our detailed response.
>
> **Large estimation bias in Fig. 5(a):** In Section 4.2, we have conducted experiments to analyze the impact of our overestimation reduction method along various penalizing constants and reported the results in Fig. 5(a). As the reviewer points out, the estimation bias may look quite large to the bare eye since we plotted the squared value of estimation bias to accurately represent the effects of overestimation and underestimation as mentioned in line 242. Regarding to reviewer's concerns for the results shown in Fig. 6, we can confirm that that discounting has been applied. We will provide additional explanation for this in the paper.
>
> [1] Kumar, Aviral, et al. "Conservative q-learning for offline reinforcement learning." Advances in Neural Information Processing Systems 33, 2020.
>
> [2] Kostrikov, Ilya, Ashvin Nair, and Sergey Levine. "Offline reinforcement learning with implicit q-learning." arXiv:2110.06169 2021.
>
> [3] An, Gaon, et al. "Uncertainty-based offline reinforcement learning with diversified q-ensemble." Advances in neural information processing systems 34, 2021.

---

> > ### Comment · Reviewer_RxmU · 2024-08-08
> > **Keep score**
> >
> > Thanks for your response, which answered my questions satisfactorily. I keep my tendency towards an accept.

---

> > > ### Author Response · Authors · 2024-08-09
> > >
> > > Thank you for the prompt response and valuable feedback. We appreciate that our response has helped clarify the issue. Please let us know if you have any additional questions or need further information.

---

### Official Review · Reviewer_36me · 2024-07-17

**Soundness:** 3
**Presentation:** 2
**Contribution:** 3
**Rating:** 7
**Confidence:** 4

**Summary:**

This paper introduces a novel approach to handling distribution shift in
off-policy RL by the means of Q-function regularization. This is
accomplished by modulating a penalty term that is overly conservative in
CQL. The authors argue that CQL overcompensates for the distribution
shift in cases where state-action pairs exist with non-negligible
density in the dataset, but are still less likely under the reference
policy than the learned policy. Based on this insight, the authors
propose to modulate the CQL value function penalty by a decreasing term
in the density of state-action pairs under the reference policy, which
activates as soon as this density crosses a threshold. Moreover, the
authors propose a prioritized sampling method to further reduce the
underestimation bias. The resulting algorithm, called EPQ, achieves
superior performance relative to its competitors across many familiar
offline RL benchmarks.

**Strengths:**

The motivation for this work was fairly strong, and the authors identified an
interesting shortcoming with the overestimation correction in CQL. Figure 1
followed by Figure 4 do a nice job of depicting the influence of this
shortcoming, and how the proposed method corrects it.

Moreover, the EPQ algorithm is deployed on a large suite of benchmarks, and
outperforms all competitors with remarkable consistency. To complement these
results, the authors conducted experiments to verify that EPQ does in fact
reduce value estimation bias, shown by comparing EPQ value estimates with
Monte-Carlo estimates, as well as predictions from CQL. The experiments are
conducted over four random seeds. While this is relatively few seeds, I think
this is acceptable given the range of tasks that were tested.

**Weaknesses:**

The presentation of the paper (e.g., writing, figures) can be improved.
In particular, many of the figures were difficult to read and/or
interpret. Both facets of Figure 2 took substantial effort to understand
for me (in fact, I think I would have more easily understood the paper
without having seen these figures; see Questions below).

The statement of Theorem 3.1 is not precise enough, particularly for the
latter claim. Moreover, I suspect there are some technical assumptions
missing; see Questions below.

Furthermore, I do not entirely understand the motivation for the
prioritized dataset. Particularly, it is not clear to me that Theorem
3.1 (the theoretical justification for EPQ) actually applies with the
prioritized dataset, since the prioritzation depends on the estimated
Q-function being updated. Beyond that, the ablation of this feature is
not very convincing.

While there is a wealth of empirical results, confidence intervals from
the baselines is largely lacking—this is especially relevant in Table 1.
See for example the `door-cloned` row: EPQ is identified as the best,
but its confidence region definitely overlaps that of CQL, and probably
those of many of the baselines as well. The same goes for
`relocate-cloned`. as well as (I'd suspect) many of the AntMaze tasks.
That said, the results do suggest that EPQ frequently outperforms its
competitors, and rarely does substantially worse.

Finally, it would have been nice to see stronger heuristics for choosing
the threshold parameter $\tau$. Figure 2a suggests that $\tau$ should be
a function of the amount of data in the dataset, but this is not
actually discussed anywhere. Rather, the authors claim to have found a
choice for $\tau$ that is inversely proportional to the volume of the
action space, but results for this choice are only given on one
environment, precluding any conclusion that this choice/trend is good in
general. The proof of Theorem 3.1 also gives a condition for determining
when $\alpha$ is large enough, which is roughly inversely proportional
to the lowest density under the reference policy over all states and
actions – therefore, choosing $\alpha$ to be inversely proportional to
the volume of the action space is only theoretically justified when the
reference policy is uniform.

## Minor Issues

The notation/definition of the Bellman operator $\mathcal{B}^\pi$ s not
exactly correct. In its definition on line 61, $\mathcal{B}^\pi$
averages over all next states $s'$. Then, in the expression on line 63,
you are taking an expectation over all state transitions $(s, a, s')$ in
the dataset with $\mathcal{B}^\pi$ evaluated in this expectation. Since
$s'$ isn't used anywhere in this expression explicitly, my assumption is
that you're using this state as the state to bootstrap from in the
application of $\mathcal{B}^\pi$; but then $\mathcal{B}^\pi$ on line 63
is not the same as its definition on line 61.

On line 83, you refer to the "actual average return $G_0$", but $G_0$
was defined as the random return (not averaged) on line 56.

Formatting of equation (2) is not nice – it almost looks like it's
depicting two separate formulas. It may read easier if you instead
colored the two factors and described their influence in the text below.

In Figure 1 and Figure 4, it would be very helpful to see where $0$ lies
on the y-axis on the estimation bias side.

Figure 2a is very busy and difficult to interpret. Firstly, I think it
would be better to highligh the magnitude of the penalty itself, as
opposed ot the penalty reduction (which implicitly depends on some
initial penalty, I'm guessing from CQL). Moreover, the relationship
between the amount of data and $\tau$ should be discussed before this
figure, even if superficially (e.g., with a sentence that says that
$\tau$ decreases as you collect more data). Then, the figure would have
a much more clear interpretation: increase the amount of data, and the
penalty will be relaxed more aggressively.

There appears to be a formatting error on line 140, "Proof) Proof…".

There is a formatting error in equation (5), $Q_(s, a)$ -\> $Q(s, a)$.

On line 243, $200k$ should be $200\mathrm{k}$.

In Table 1, the "total" rows are not good indications of performance,
firstly because the returns for the different environments are not
normalized. That said, the results for EPQ still look good if you
neglect the "total" rows.

In Figure 5, it would help to use a different line style to emphasize
which curve corresponds to EPQ. I found it difficult to distinguish EPQ
from CQL ($\alpha=0.0$) – fortunately these curves generally occupied
disjoint regions in the graphs.

**Questions:**

In Figure 1, what is the relationship between $\tau_i$ and $N_i$
($i=1,2$)? Such a relationship has not been discussed up to this point.

In the proof of Theorem 3.1, I believe some assumptions are missing. If
$\pi(\cdot\mid
s)$ is ever supported on an action $a'$ such that
$\hat{\beta}(a'\mid s) = 0$, then $\Delta^\pi_{EPQ}\to\infty$.
Therefore, there would be no $\alpha$ large enough to underestimate the
value function for your argument on line 465 as long as there exists a
single $(s, a)$ in the dataset for which $\xi^\delta(s, a) > 0$. Since
$\hat{\beta}$ was defined to be the empirical conditional distribution
over actions from the dataset, this result actually suggests that
$\alpha$ must be infinite whenever your dataset does not fully cover the
action space (which is always the case in the experiments, where the
action space is continuous).

In table 1, why aren't confidence intervals given for the baseline
methods?

Why are there no confidence regions shown in Figure 6? Particularly, it
would have been helpful to see these in Figure 6a. As it stands, the
effect of the prioritized dataset depicted in this figure is a little
underwhelming.

With regard to the analysis of the penalty threshold, why should we
scale $\tau$ linearly with the density of the uniform distribution over
$\mathcal{A}$ (that is, inversely proportional to the volume of the
action space)? Figure 6b does not indicate whether the choice of
$\tau = 0.2\rho$ is actually a good choice across environments, so
indeed it could have just been that $\tau=0.2\rho$ happens to work well
in `hopper-medium` by chance. You have have investigated (or at least
presented the results) that show how this form of scaling with the
action space performs.

**Limitations:**

Limitations are mostly discussed, except for potentially missing assumptions in
Theorem 3.1.

---

> ### Author Rebuttal · Authors · 2024-08-06
>
> Thank you for your valuable feedback. We have addressed the feedback regarding Fig. 2 and conducted additional ablation studies to answer the reviewer's concerns related to the proposed prioritized dataset (PD) and the penalty control threshold $\tau$, as detailed in our global response. In addition, we have included confidence intervals for all results in the author-provided PDF.
>
> **Scaling and selection of the threshold $\tau$**
>
> As mentioned in the global response, the penalty control threshold $\tau$ is proportional to the log-density of $\mathrm{Unif}(\mathcal{A})$. However, we did not intend for the threshold $\tau$ to scale proportionally to the action volume. Instead, since $\tau$ is compared to the log-density of $\hat{\beta}$, a distribution over the action space $\mathcal{A}$, we designed the setup so that similar scale factors $\tau/\rho \in [0.2,...,10]$ yield consistent thresholding effects across different action dimensions, without being overly sensitive to changes in the action space.
>
> Based on the experimental results provided for the global response, the overestimation error arises from samples that have not been visited, so the optimal $\tau$ tends to be determined by how diversely actions are sampled in the dataset for each environment. Fig. 2 and R.1 suggest that, "with a fixed data distribution", as the number of datasets $N$ increases, actions are sampled more diversely, which implies that $\tau$ should be lower. In actual experimental environments, however, since the data distribution varies across environments and states, these factors need to be additionally considered for choosing the optimal $\tau$. According to our analysis related to $\tau$, a low threshold is effective when the dataset contains sufficiently diverse state-action pairs, whereas a higher threshold is preferable when the dataset is less comprehensive.
>
> **Confidence interval of the experimental results**
>
> In offline setups, where performance measurement methods are generally similar, it is common to cite reported results for certain algorithms or experiments [1, 2, 3]. As outlined in Section 4.1, we utilized reported results from the recent MISA paper [1], which did not include standard deviations. For the missing results, we conducted our own experiments to provide the necessary performance data. However, for the experiments we reproduced, such as the modified CQL comparisons in Table 6 for Adroit tasks, we have included confidence intervals as requested by the reviewer. Even when considering these confidence intervals, our proposed EPQ algorithm consistently demonstrates superior performance compared to the CQL baseline. We believe that the proposed EPQ's superiority is well-supported, given its significant enhancement in average performance relative to other baselines.
>
> **Theoretical soundness of Theorem 3.1**
>
> **Penalizing constant $\alpha$:** Theorem 3.1 states that when the penalizing constant $\alpha$ is sufficiently large, the $Q$-values $\hat{Q}^\pi$ learned based on the EPQ penalty $\mathcal{P}\_\tau$ will underestimate the true $Q$-values. The reviewer inquired whether, according to Theorem 3.1, $\alpha$ needs to be infinite for $\hat{\beta}$ approaching zero to allow for underestimation. We observed that this confusion arises from a typo in the proof of Theorem 3.1 in Appendix A.1. Currently, the proof incorrectly states that $\alpha$ must satisfy $\alpha \geq \max_{s,a \in D}[\xi^\delta(s,a)] \cdot \max_{s \in D} \Delta_{EPQ}^\pi(s)$. However, the correct condition should be $\alpha \geq \max_{s,a \in D}[\xi^\delta(s,a)] \cdot \max_{s \in D} (\Delta_{EPQ}^\pi(s))^{-1}$. As $\hat{\beta}$ approaches zero, $\Delta_{EPQ}^\pi(s)$increases towards infinity, allowing the condition to still be satisfied with a relatively small $\alpha$, which is an intuitively natural result. This behavior is empirically supported by our experiments, as shown in Fig. 4(c), where we observe significant underestimation of the $Q$-values when $\hat{\beta}$ is very small within the support of $\pi$. Thank you for pointing out this issue, and we will correct the typo in the proof to clarify this matter.
>
> **Theorem 3.1 with PD:** As the reviewer noted, Theorem 3.1 assumes a scenario without the PD and provides a proof under that assumption. When considering PD, $\hat{\beta}^Q$ continuously changes with $Q$, which makes it challenging for Theorem 3.1 to hold directly. However, if we assume a fixed $Q$ for PD, Theorem 3.1 can be applied to a fixed $\hat{\beta}^Q$ in a similar manner. To address the reviewer's concern in the presence of PD, we use the actual returns from the dataset to obtain $\hat{\beta}^Q$ rather than the learned $Q$ during training, as discussed in Appendix B.4. We will add a more detailed explanation of this part in the main paper.
>
> **Minor issues**
>
> We agree that our notation of the Bellman operator $\mathcal{B}^\pi$ and the return $G_0$ should be changed as the reviewer suggests. For $\mathcal{B}^\pi$, since we are not using the next state $s'$ the expectation should be over $(s,a)$ rather than over all state transitions $(s,a,s')$. For $G_0$ we will address the issue of overlapping notation between the "random return" defined in line 56 and the "actual average return" referenced in line 83, as well as correct any typos. Additionally, we will incorporate the reviewer's suggestions on formatting to improve the clarity of the paper. We appreciate your valuable feedback and will work to enhance the paper based on the reviewer's suggestions.
>
> [1] Ma, Xiao, et al. "Mutual information regularized offline reinforcement learning." Advances in Neural Information Processing Systems 36, 2024.
>
> [2] Kostrikov, Ilya, Ashvin Nair, and Sergey Levine. "Offline reinforcement learning with implicit q-learning." arXiv preprint arXiv:2110.06169, 2021.
>
> [3] An, Gaon, et al. "Uncertainty-based offline reinforcement learning with diversified q-ensemble." Advances in neural information processing systems 34, 2021.

---

> > ### Comment · Reviewer_36me · 2024-08-08
> >
> > Thanks to the authors for the detailed response.
> >
> > **Choice of $\tau$**: Thanks a lot for the discussion and the additional experimental data. This is helpful, and this (in my opinion) is a more useful argument / advice for choosing this parameter than what was originally stated in the submission.
> >
> > **Confidence intervals**: Thanks again for the additional experimental data, I am satisfied with these results now.
> >
> > **Penalizing constant $\alpha$**: I think I see what you mean, though it would be helpful maybe if you could write out this deduction explicitly. For example, explicitly show why $\alpha \geq \mathsf{corrected bound}$ leads to underestimation.

---

> > > ### Author Response · Authors · 2024-08-09
> > >
> > > Thank you for the prompt response from the reviewer. To provide a more intuitive understanding of penalizing the constant $\alpha$, as requested by the reviewer, we will explicitly explain below how the value function of EPQ underestimates the true value.
> > >
> > > **Explicit derivation of Theorem 3.1 to show underestimation**
> > >
> > > We start from line 462 in Appendix A.1, which states that when $V_{k+1}$ converges to $V_\infty$, then the converged value function $V_\infty$ of EPQ satisfies
> > >
> > > $V_\infty(s)  = V^\pi(s) + (I-\gamma P^\pi)^{-1}\cdot$ \{ $- \alpha\Delta_{EPQ}^{\pi}(s) + \mathbb{E}_{a\sim\pi}[\xi^\delta(s,a)]$\},
> > >
> > > where $\xi^\delta(s,a)$ and $\Delta_{EPQ}^{\pi}(s)$ are positive $\forall~ s,a$, assuming $\pi\neq\hat{\beta}$. (if $\pi=\hat{\beta}$, then there will be no overestimation error.)
> > >
> > >
> > > If we choose the penalizing constant $\alpha$ that satisfies $\alpha \geq \max_{s,a\in D}[\xi^\delta(s,a)]\cdot\max_{s\in D} (\Delta_{EPQ}^\pi(s))^{-1}$, then
> > >
> > > $- \alpha\cdot\Delta\_{EPQ}^{\pi}(s) + \mathbb{E}\_{a\sim\pi}[\xi^\delta(s,a)] $
> > >
> > > $\leq- \max\_{s,a\in D}[\xi^\delta(s,a)]\cdot \underbrace{\max\_{s\in D} (\Delta\_{EPQ}^\pi(s))^{-1} \cdot \Delta\_{EPQ}^{\pi}(s)}_{\geq 1} +  \mathbb{E}\_{a\sim\pi}[\xi^\delta(s,a)]$
> > >
> > > $\leq- \max\_{s,a\in D}[\xi^\delta(s,a)] +  \mathbb{E}\_{a\sim\pi}[\xi^\delta(s,a)] \leq 0,~~~~\forall s,$
> > >
> > > Since $I-\gamma P^\pi$ is non-singular $M$-matrix and the inverse of non-singular $M$-matrix is non-negative (Please see "M-matrix" in Wikipedia), i.e., all elements of $(I - \gamma P^\pi)^{-1}$ are non-negative, then
> > >
> > > $V_\infty(s)  = V^\pi(s) + (I-\gamma P^\pi)^{-1}\cdot$ \{ $- \alpha\Delta\_{EPQ}^{\pi}(s) + \mathbb{E}\_{a\sim\pi}[\xi^\delta(s,a)]$ \}$\leq V^\pi(s),~\forall s$.
> > >
> > > Thus, we can conclude that $V_\infty$ of EPQ underestimates the true value $V^\pi$. We hope this explanation helps clarify the concept of Theorem 3.1.

---

> > > > ### Comment · Reviewer_36me · 2024-08-09
> > > >
> > > > Thanks a lot for the clarification, I will raise my score.

---

> > > > > ### Author Response · Authors · 2024-08-12
> > > > >
> > > > > Thank you for your thoughtful response. We will make sure to incorporate the points you mentioned into the paper. If you have any additional questions or need further information, please let us know.

---

### Author Rebuttal · Authors · 2024-08-06

Thank you for your valuable feedback. Based on the reviewers' comments, many expressed difficulties in understanding the figures presented in the paper and suggested that additional ablation studies would be beneficial. Therefore, we provide the following detailed responses to address each of these points:


**Clarity issues for figures in the paper**


In particular, there seems to be difficulty in understanding Fig. 2, which illustrates the motivation behind EPQ, as it lacks sufficient detail. To provide a clearer illustration of our methods, we have revised Fig. 2(a) and Fig. 2(b) into Fig. R.1 and Fig. R.2, respectively, which are included in the author-provided one-page PDF.

**Fig. R.1:** As detailed in Section 3.2, our exclusive penalty is designed to minimize unnecessary bias in the $Q$-function by imposing penalties only when the policy actions are insufficiently represented in the dataset. To illustrate the rationale behind our exclusive penalty, Fig. R.1(a) depicts the log-probability of $\hat{\beta}$ and the thresholds $\tau$ used for penalty adaptation, with $N$ representing the number of data points. In Fig. R.1(a), if the log-probability $\log\hat{\beta}$ of an action $a \in \mathcal{A}$ exceeds the threshold $\tau$, this indicates that the action $a$ is sufficiently represented in the dataset, thus, we reduce the penalty for such actions. Furthermore, as shown in Fig. R.1, when the number of actions increases from $N_1$ to $N_2$, the threshold for determining "enough data" decreases from $\tau_1$ to $\tau_2$, even if the data distribution remains unchanged.

Fig. R.1(b) illustrates the proposed penalty adaptation factor $f\_\tau^{\pi,\hat{\beta}} = \mathbb{E}\_{\pi}[x\_\tau^{\hat{\beta}}]$ for a given $\hat{\beta}$ and policy $\pi$. Here, $x_\tau^{\hat{\beta}} = \min(1.0, \exp(-(\log \hat{\beta} - \tau)))$ represents the amount of adaptive penalty that is reduced as $\log \hat{\beta}$ exceeds the threshold $\tau$. In Fig. R.1(b), $x_{\tau_1}^{\hat{\beta}}$ is larger than $x_{\tau_2}^{\hat{\beta}}$ because $\tau_1 > \tau_2$. Thus, the adaptation factor $f_\tau^{\pi,\hat{\beta}}$ indicates the average penalty that policy actions should receive. As illustrated in Fig. R.1(b), the adaptation factors for different policies vary with their position. Specifically, for threshold $\tau_1$, we have $f_{\tau_1}^{\pi_1,\hat{\beta}} = f_{\tau_1}^{\pi_2,\hat{\beta}} = 1$ and $f_{\tau_1}^{\pi_3,\hat{\beta}} < 1$. For threshold $\tau_2$, $f_{\tau_2}^{\pi_3,\hat{\beta}} < f_{\tau_2}^{\pi_1,\hat{\beta}} < f_{\tau_2}^{\pi_2,\hat{\beta}} < 1$, as depicted in Fig. R.1(b).

**Fig. R.2:** As explained in Section 3.3, we introduce the prioritized dataset (PD) to further reduce the penalty when the policy is highly concentrated on actions that maximize the $Q$-function. To illustrate this, Fig. R.2(a) shows the difference between the original data distribution $\hat{\beta}$ and the modified data distribution $\hat{\beta}^Q$ after applying PD, and Fig. R.2(b) depicts the corresponding penalty graphs. As shown in Fig. R.2(a), when the policy $\pi$ focuses on specific actions, the penalty $\frac{\pi}{\hat{\beta}} - 1$ increases significantly in Fig. R.2(b). In contrast, by applying PD, $\hat{\beta}$ is adjusted to approach $\hat{\beta}^Q \propto \beta \exp(Q)$, aligning the data distribution more closely with the policy $\pi$. Consequently, the penalty is substantially reduced, as depicted in Fig. R.2(b). We believe that Fig. R.1 and Fig. R.2 will provide a clearer understanding of our proposed methods.

**Additional ablation studies**

In response to the feedback from reviewers 36me and Hh4A, we have conducted additional ablation studies for the Hopper-random, Hopper-medium, and HalfCheetah-medium tasks. These tasks demonstrate a significant performance improvement of our method compared to the baseline CQL, as discussed in Section 4.2. Figure R.3(a) provides a component evaluation to illustrate the impact of the proposed PD, while Figure R.3(b) examines performance across various penalty control thresholds $\tau \in [0.2\rho, 0.5\rho, 1.0\rho, 2.0\rho, 5.0\rho, 10.0\rho]$, where $\rho$ represents the log-density of $\textrm{Unif}(\mathcal{A})$. (There is a typo in the paper. We will fix it.) Note that $\rho$ is negative, so $\tau = 10\rho$ is the lowest threshold and $\tau = 0.2\rho$ is the highest.

**The effectiveness of PD:** For the component evaluation, reviewers expressed concerns about the effectiveness of the proposed PD, particularly noting its minimal impact on performance in the Hopper-medium task. However, as shown in Fig. R.3(a), while PD has a limited effect on the Hopper-random and Hopper-medium tasks, it significantly improves performance in the HalfCheetah-medium task, thereby validating the effectiveness of PD.

**The analysis of threshold $\tau$:** Additionally, reviewers inquired about the selection and impact of the proposed hyperparameter $\tau$. Fig. R.3(b) provides insights into the optimal $\tau$ values for each task. The results indicate that in tasks like Hopper-medium, where a variety of actions are not sufficiently sampled, a higher threshold performs better. Conversely, in tasks like Hopper-random, where a broad range of actions is sampled, a lower threshold is more effective. An exception is the HalfCheetah-medium task, which, despite having fewer action variations, visits a diverse range of states. This results in lower overestimation errors for out-of-distribution actions, benefiting from a lower threshold. Additionally, the Adroit task performs well with an extremely high threshold due to minimal noise in the dataset, leading to a limited variety of state-action pairs.

We believe that the additional experiments and analyses, as suggested by the reviewers, robustly validate the effectiveness of the proposed components and significantly enhance the quality of the paper. We again appreciate the reviewers' valuable feedback.

---

### Decision · Program_Chairs · 2024-09-25

**Decision:**

Accept (spotlight)

**Comment:**

All reviewers (and myself) are in agreement that this work is well-motivated and provides enough empirical evidence for the introduced method. As such, I am recommending acceptance.